# What Truly Matters in Trajectory Prediction for Autonomous Driving?

**Phong Tran**[1*]    **Haoran Wu**[1,2*]    **Cunjun Yu**[1*]  **Panpan Cai**[3]  **Sifa Zheng**[2]   **David Hsu**[1]

[1] National University of Singapore
[2] Tsinghua University
[3] Shanghai Jiao Tong University

## Abstract

Trajectory prediction plays a vital role in the performance of autonomous driving systems, and prediction accuracy, such as average displacement error (ADE) or final displacement error (FDE), is widely used as a performance metric. However, a significant disparity exists between the accuracy of predictors on fixed datasets and driving performance when the predictors are used downstream for vehicle control, because of a *dynamics gap*. In the real world, the prediction algorithm influences the behavior of the ego vehicle, which, in turn, influences the behaviors of other vehicles nearby. This interaction results in predictor-specific dynamics that directly impacts prediction results. In fixed datasets, since other vehicles' responses are predetermined, this interaction effect is lost, leading to a significant dynamics gap. This paper studies the overlooked significance of this dynamics gap. We also examine several other factors contributing to the disparity between prediction performance and driving performance. The findings highlight the trade-off between the predictor's computational efficiency and prediction accuracy in determining real-world driving performance. In summary, an *interactive*, *task-driven* evaluation protocol for trajectory prediction is crucial to capture its effectiveness for autonomous driving. Source code along with experimental settings is available online.

## 1   Introduction

Current trajectory prediction evaluation [29, 9, 5] relies on real-world datasets, operating under the assumption that dataset accuracy is equivalent to prediction capability. We refer to this as *Static Evaluation*. This methodology, however, falls short when the predictor serves as a sub-module for downstream tasks in Autonomous Driving (AD) [28, 24]. As illustrated in Figure 1, the static evaluation metrics on datasets, such as Average Displacement Error (ADE) and Final Displacement Error (FDE), do not necessarily reflect the actual driving performance [36, 8, 10]. In contrast to the conventional focus on uncertainty [39, 16] and potential interaction [27], we demonstrate that this disparity stems from the overlooked dynamics gap between fixed datasets and AD systems and the computational efficiency of predictors. The trade-off between these two factors matters in the actual prediction performance within the entire AD system.

The dynamics gap arises from the fact that the behavior of the autonomous vehicle, also known as the ego-agent, varies with different trajectory predictors, as presented in Figure 2. In real-world scenarios, the ego-agent utilizes predictors to determine its actions. Different predictors result in varied behaviors of the ego-agent, which, in turn, influence the future behaviors of other road users, leading to different dynamics within the environment. This directly affects the accuracy of predictions, as other agents behave differently. Since the ego-agent's actions are predetermined on

---

*Equal contribution.

37th Conference on Neural Information Processing Systems (NeurIPS 2023).

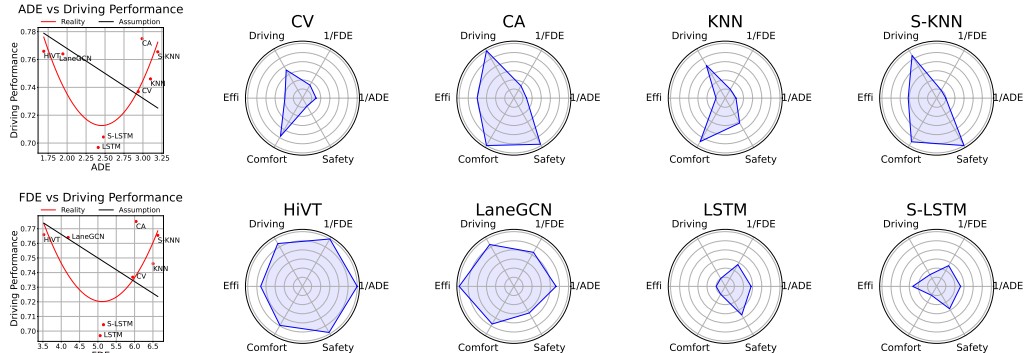

Figure 1: Prediction accuracy *vs* driving performance. Contrary to popular belief (left, black curves), our study indicates no strong correlation between common prediction evaluation metrics and driving performance (left, red curves). Eight representative prediction models are selected: Constant Velocity (CV) [30], Constant Acceleration (CA) [30], K-Nearest Neighbor (KNN) [9], Social KNN (S-KNN) [9], HiVT [40], LaneGCN [22], LSTM, and Social LSTM (S-LSTM) [2]. The definition of driving performance metrics can be found in Section 4.4. Details can be found in the supplementary materials.

the dataset, there exists a significant disparity between the dynamics represented in the dataset and the actual driving scenario when evaluating a specific trajectory predictor. To tackle this issue, we propose the use of an interactive simulation environment to evaluate the predictor for downstream decision-making. This environment enables *Dynamic Evaluation* as the ego-agent operates with the specific predictor, thus, mitigating the dynamics gap. We demonstrate a strong correlation between the dynamic evaluation metrics and driving performance through extensive experiments. This underscores the dominant role of the dynamics gap in aligning prediction accuracy with driving performance and emphasizes the importance of incorporating it into the evaluation process.

The awareness of the dynamics gap significantly enhances the correlation between prediction accuracy and driving performance. However, there are still factors that affect driving performance apart from prediction accuracy. The downstream modules in AD systems, with different tasks and varying levels of complexity, impose different requirements on prediction models. In our experiments, we vary the type of planner and the time constraint for the motion planning task. Our findings suggest that predictors' computational efficiency plays a vital role in real-time tasks. Moreover, it is the trade-off between prediction accuracy and computational efficiency that determines driving performance. This highlights the necessity for task-driven prediction evaluation.

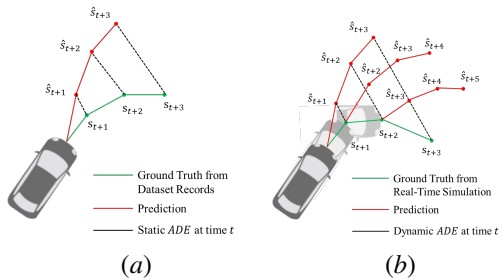

Figure 2: Dynamics Gap. (*a*) In static evaluation, the agent's motion is determined and unaffected by predictors. (*b*) In the real-world, different predictors result in varied behaviors of the agent, which directly affects the ground truth of prediction.

In this paper, we aim to address two specific aspects of trajectory prediction for AD systems. Firstly, we uncover the limitation of static prediction evaluation systems in accurately reflecting driving performance. We demonstrate the dominant role of the dynamics gap in causing the disparity. Secondly, we emphasize the necessity of an interactive, task-driven evaluation protocol that incorporates the trade-off between dynamic prediction accuracy and computational efficiency. The protocol presents a promising way for applying prediction models in real-world autonomous driving, which is the truly mattered aspect in trajectory prediction for autonomous driving.

## 2 Related Work

### 2.1 Motion Prediction and Evaluation

Motion prediction methods can be classified along three dimensions [17]: modeling approach, output type, and situational awareness. The modeling approach includes physics-based models [30, 3]

that use physics to simulate agents' forward motion, and learning-based models [40, 22] that learn and predict motion patterns from data. The output type can be intention, single-trajectory [26], multi-trajectory [18], or occupancy map [14, 15]. These outputs differ in the type of motion they predict and how they handle the uncertainty of future states. The situational awareness includes unawareness, interaction [2], scene, and map awareness [31]. It refers to the predictor's ability to incorporate environmental information, which is crucial for collision avoidance and efficient driving. Most researchers [29, 25] and competitions [9, 5] evaluate the performance of prediction models on real-world datasets, in which ADE/FDE and their probabilistic variants minADE/minFDE are commonly used metrics. However, these metrics fail to capture the dynamics gap between datasets and real-world scenarios, as the actions of the ego-agent remain unaffected by predictors. In this study, we select four model-based and six learning-based models with varying output types and situational awareness to cover a wide range of prediction models. We implement these predictors in both datasets and an interactive simulation environment to illustrate the limitation of current prediction evaluation in accurately reflecting driving performance, due to the neglect of the dynamics gap.

## 2.2 Task-aware Motion Prediction and Evaluation

Task-aware motion prediction remains an underexplored area in research. While some studies touch upon the subject, they focus on proposing task-aware metrics for training or eliminating improper predictions on datasets. One notable example of training on task-aware metrics is the Planning KL Divergence (PKL) metric [28]. Although designed for 3D object detection, it measures the similarity between detection and ground truth by calculating the difference in ego-plan performance. In the context of motion prediction, Rowan et al. [24] propose a control-aware metric (CAPO) similar to PKL. CAPO employs attention [35] to find correlations between predicted trajectories, assigning higher weights to agents inducing more significant reactions. Similarly, the Task-Informed method [19] assumes a set of candidate trajectories from the planner and adds the training loss for each candidate. Another line of work focuses on designing task-aware functions to eliminate improper predictions [21, 13]. The proposed metric can capture unrealistic predictions and better correlate with driving performance in planner-agnostic settings. However, these works are conducted in an open-loop manner, neglecting the disparity between open-loop evaluation and real-world driving performance. To the best of our knowledge, we are the first to conduct a thorough investigation into the disparity and demonstrate the dominant role of the dynamics gap on this issue.

## 3 Planning with Motion Prediction

### 3.1 Problem Formulation

In this section, we present the problem formulation for motion prediction with planning in the context of autonomous driving. Different from conventional formulations that consider the planner as a policy, our formulation employs the predictor as the policy. This enables us to identify the primary challenge in developing predictors for real-world autonomous driving. We define the prediction problem as an MDP: $M = (\mathcal{S}, \mathcal{A}, \mathcal{T}, \mathcal{R})$.

**State.** The state of the system is represented by the tuple $s = (s^{ego}, s^{exo})$ for $s \in \mathcal{S}$, where $s^{ego}$ denotes the state of the AV including historical information, $s^{exo}$ denotes that of surrounding traffic participants. Specifically, $s^{ego} = \{s^{ego}_{t-t_{obs}}, \ldots, s^{ego}_t\}$, where $t_{obs}$ represents the observation horizon and $t$ denotes the current timestep. The state for each vehicle includes its position, velocity, heading, and other relevant attributes, e.g., $s^{ego}_t = (x^{ego}_t, y^{ego}_t, v^{ego}_t, \theta^{ego}_t)$.

**Action.** At each time step, the AV can take an action $a \in \mathcal{A}$, where $\mathcal{A}$ is the action space. We denote the $n$ surrounding traffic participants as $i \in \{1, \ldots, n\}$. The action consists of the AV's prediction of all exo-agents, e.g., $a = \{a^1, \ldots, a^n\}$.

**Transition Function.** The transition function $\mathcal{T} : \mathcal{S} \times \mathcal{A} \to \mathcal{S}$ defines the dynamics of the system and how the system evolves as a result of the action of the ego vehicle. In the MDP, it refers to: $T(s_{t+1}|s_t, a_t) = T_1(s^{ego}_{t+1}|s_t, a_t)T_2(s^{exo}_{t+1}|s_t), T \in \mathcal{T}$. We denote the planner as a mapping function $T_1$ that takes the current state $s_t$ and action $a_t$ as input and outputs the AV's next state $s^{ego}_{t+1}$. The motion models of exo-agents are integrated into the mapping function $T_2$, which takes the current state $s_t$ as input and generates the next state $s^{exo}_{t+1}$ for all exo-agents.

**Predictor.** The predictor is represented as a policy $a = \pi(s)$ that takes the current state $s$ as input, and produces an action $a \in \mathcal{A}$ as output.

**Objective Function.** The objective of prediction is to accurately predict exo-agents future states while considering their interactions and map information. The reward function $R : \mathcal{S} \times \mathcal{A} \rightarrow \mathbb{R}$ maps a state-action pair to a real-valued reward. The objective function is the accumulated reward over the task horizon $H$, defined as $J(s, a) = \sum_{t=0}^{H} R(s_t, a_t)$.

**Optimal Policy.** The objective of autonomous driving is to identify the optimal policy that minimizes the objective function under the real-world transition function. The optimal policy $\pi^*$ is expressed as: $\pi^* = \underset{T_1 = T_1^*, T_2 = T_2^*}{\arg\max} \mathbb{E}_\pi[\sum_{t=0}^{H} R(s_t, a_t)]$, where $T_1^*$ represents any realistic planners used in AVs, and $T_2^*$ denotes the integrated motion model of exo-agents in the real world.

## 3.2 The Limitation of Static Evaluation

Traditional prediction methods overlook the difference between the transition function $T_1^*$ in real-world autonomous driving and the represented $\hat{T}_1$ in datasets. These methods focus on training a policy using well-designed reward functions to perform effectively within the state distribution of datasets. The trained policy $\pi$ satisfies:

$$\pi = \underset{T_1 = \hat{T}_1, T_2 = T_2^*}{\arg\max} \mathbb{E}_\pi[\sum_{t=0}^{H} R(s_t, a_t)]. \tag{1}$$

In this context, $\hat{T}_1$ serves as a static planner that remains unaffected by predictors. Regardless of the input, the static planner $\hat{T}_1$ generates the recorded next state. As a result, the mapping function $T_2^*$ remains consistent with the real world since the input state is unchanged. We define the disparity in future states resulting from changes in the ego-planner as the dynamics gap in static evaluation, denoted as $G = T_1^* - \hat{T}_1$.

The proposed planning-aware metrics, such as PKL [28] and CAPO [24], aimed to address the disparity between prediction accuracy and the ultimate driving performance by improving the cost function. However, unless the dynamics gap is mitigated, the optimization deviation plays a dominant role in causing the disparity and should be addressed as a priority.

## 3.3 Measuring the Impact of Dynamics Gap

Conducting real-world tests is the most effective way to address and assess the impact of the dynamics gap. However, due to its high cost, the simulator serves as a proxy. The ego-agent utilizes the realistic planner $T_1^*$ and the motion model $T_{sim}^*$ of the simulator. The optimal policy $\pi_{sim}^*$ satisfies:

$$\pi_{sim}^* = \underset{T_1 = T_1^*, T_2 = T_2^{sim}}{\arg\max} \mathbb{E}_\pi[\sum_{t=0}^{H} R(s_t, a_t)]. \tag{2}$$

To evaluate the impact of the dynamics gap, we train various predictors on the Alignment dataset obtained from the simulator, the trained policy $\pi_{sim}$ satisfies:

$$\pi_{sim} = \underset{T_1 = \hat{T}_1, T_2 = T_2^{sim}}{\arg\max} \mathbb{E}_\pi[\sum_{t=0}^{H} R(s_t, a_t)]. \tag{3}$$

The equations suggest that the dynamics gap between the Alignment dataset and simulator mirrors the dynamics gap between the real-world dataset and actual autonomous driving scenarios. Thus, it is reasonable to employ the simulator for evaluating the influence of the dynamics gap. However, the simulator merely serves as a substitute to mitigate the dynamics gap, leaving a remaining dynamics gap represented by $G' = T_2^* - T_2^{sim}$.

## 4 Experimental Setup

Our aim is to identify the key factors involved in trajectory prediction for autonomous driving and recognize their impact on driving performance by simulating real-world scenarios that vehicles might encounter. The ultimate goal is to introduce an interactive, task-driven prediction evaluation protocol. To achieve this objective, we need to determine four crucial components: *1*) motion prediction methods to be covered; *2*) realistic planners to employ prediction models; *3*) the simulator to replicate interactive scenarios; *4*) evaluation metrics to assess the effectiveness of the key factors involved in motion prediction with respect to real-world driving performance.

Table 1: Selected prediction methods.

| Modeling Approach | Method | Output Type | Interaction Aware | Scene Aware | Map Aware |
|---|---|---|---|---|---|
| Model-based | CV [30] | ST | ✗ | ✗ | ✗ |
| | CA [30] | ST | ✗ | ✗ | ✗ |
| | KNN [9] | MT | ✗ | ✗ | ✗ |
| | S-KNN [9] | MT | ✓ | ✗ | ✗ |
| Data-driven | LSTM | ST | ✗ | ✗ | ✗ |
| | S-LSTM [2] | ST | ✓ | ✗ | ✗ |
| | HiVT [40] | MT | ✓ | ✓ | ✓ |
| | LaneGCN [22] | MT | ✓ | ✓ | ✓ |
| | HOME [14] | OM | ✓ | ✓ | ✓ |
| | DSP [38] | MT | ✓ | ✓ | ✓ |

*Abbreviations: **ST**: Single-Trajectory, **MT**: Multi-Trajectory, **OM**: Occupancy Map.

## 4.1 Motion Prediction Methods

We select 10 representative prediction models to achieve comprehensive coverage of mainstream approaches, ranging from simple model-based methods to complex data-driven approaches, as presented in Table 1. Constant Velocity (CV) and Constant Acceleration (CA) [30] assume that the predicted agent maintains a constant speed or acceleration within the prediction horizon. K-Nearest Neighbor (KNN) predicts an agent's future trajectory based on most similar trajectories, while Social-KNN (S-KNN) [9] extends it by also considering the similarity of surrounding agents. These methods are widely used as baselines given their widespread effectiveness in simple prediction cases. Social LSTM (S-LSTM) [2], HiVT [40], LaneGCN [22], and HOME [14] represent four distinct types of neural networks: RNN, Transformer, GNN, and CNN. DSP [38] utilizes a hybrid design of neural networks, representing state-of-the-art prediction models.

## 4.2 Planners

An ideal planner should: *1*) be able to handle state and action uncertainty; *2*) take into account multiple factors of driving performance, such as safety (collision avoidance), efficiency (timely goal achievement), and comfort (smooth driving); *3*) be aware of interactions with other agents; and *4*) supports real-time execution. We select two realistic planners based on these criteria: a simplistic planner that only meets *2*) and *4*), and a sophisticated planner that satisfies all of *1*) - *4*). This allows us to draw planner-agnostic conclusions.

**RVO.** The RVO planner [34] is a simplistic planner that solves the optimization problem in the velocity space under collision avoidance constraints. The planner uses motion predictions to avoid possible collisions with the deterministic motions, and thus does not consider the state and action uncertainty. As the RVO planner does not maintain states between consecutive timesteps, it also cannot optimize its planning with respect to interactions with other agents. The objective function of the RVO planner involves safety and efficiency within a short time window, and the RVO planner executes in real time.

**DESPOT.** The DESPOT planner [32] is a state-of-the-art belief-space planning algorithm that address uncertainties near-optimally. To account for stochastic states and actions, we adopt the bicycle model, a kinematic model with two degrees of freedom, and introduce Gaussian noise to the displacement. DESPOT considers the system state, context information, and the ego-agent's action to predict the future states of exo-agents. This allows it to consider the interaction between agents. Moreover, the objective function of DESPOT incorporates safety, efficiency, and comfort metrics, making it an ideal algorithm for planning in complex and dynamic environments.

We use these planners to control the speed of the ego-agent while pure-pursuit algorithm [11] for adjusting the steering angle. Details can be found in the supplementary materials.

## 4.3 Simulator

In order to evaluate different prediction models, the ideal simulator should: *1*) provide real-world maps and agents; *2*) model potential unregulated behaviors; *3*) accurately mirror the interactions between agents; and *4*) provide realistic perception data for effective planning. We choose the SUMMIT simulator [6] for our experiments since it meets all the criteria, as demonstrated in Table 2. SUMMIT is a sophisticated simulator based on the Carla [12] framework, offering various real-world

Table 2: Comparison between simulators.

| Simulator | Real-World Maps | Unregulated Behaviors | Dense Interactions | Realistic Visuals |
|---|---|---|---|---|
| SUMO [23] | ✓ | ✗ | ✓ | ✗ |
| TrafficSim [33] | ✗ | ✓ | ✓ | ✗ |
| TORCS [7] | ✗ | ✓ | ✗ | ✓ |
| BARK [4] | ✗ | ✓ | ✗ | ✗ |
| Apollo [1] | ✗ | ✗ | ✗ | ✗ |
| Symphony [20] | ✓ | ✓ | ✗ | ✓ |
| SUMMIT [6] | ✓ | ✓ | ✓ | ✓ |

maps and agents to create diverse and challenging scenarios. It uses a realistic motion model to simulate interactions between agents and supports the simulation of crowded scenes, complex traffic conditions, and unregulated behaviors.

There are two distinct concepts of time in our experiments: *simulation time* and *real time*. The former corresponds to the duration of actions in the simulator, while the latter represents the wall time consumed by the planner. By default, the simulator runs in asynchronous mode. In this mode, the simulator and planner run individually, and the simulation time is equal to the real time. However, in this study, we employ the SUMMIT simulator in synchronous mode to accommodate slow predictors. In this mode, the simulator waits for the planner to establish communication before proceeding to the next step. Since the simulation time for each execution step remains constant at $0.03$ s, the ratio between simulation time and real time can be manually set by varying the communication frequency, known as the tick rate. For example, once the tick rate is set to 30 Hz, the simulation time equals the real time. But, when the tick rate is set to 3 Hz, the ratio between simulation time and real time is 0.1.

## 4.4 Evaluation Protocols

**Motion Prediction Performance Metrics.** Four commonly used prediction performance metrics are employed in this study, as presented in Table 3. In our experiments, the consensus of K=6 is adopted. While ADE/FDE can be applied to evaluate single-trajectory prediction models, their probabilistic variants minADE and minFDE can be applied to evaluate multi-trajectory predictors.

**Driving Performance Metrics.** The driving performance is primarily determined by three factors: safety, comfort, and efficiency. Let $H$ represent the total timestep for each scenario.

Safety is typically evaluated in terms of the collision rate. The collision is determined whether the smallest distance between the ego-agent and surrounding agents is smaller than a threshold $\epsilon$, that is: $P_{\text{Safety}} = \frac{1}{H} \sum_{t=1}^{H} \mathbb{I}[\min\{||s_A^t - s|| : s \in \{s_1^t, ..., s_n^t\}\} < \epsilon]$, where $|| \cdot ||$ is the L2 distance between the ego-agent's bounding box and the exo-agent's bounding box, $\mathbb{I}$ is the boolean indicator function. We set $\epsilon = 1$m for our experiments since the DESPOT planner rarely causes real collisions.

Efficiency is evaluated by the average speed of the ego-agent over the whole scenario: $P_{\text{Efficiency}} = \frac{1}{H} \sum_{t=1}^{H} v_A^t$. Comfort is represented by the jerk of the ego-agent, which is the rate of change of acceleration with respect to time: $P_{\text{Comfort}} = \frac{1}{H} \sum_{t=1}^{H} \dddot{v}_A^t$.

Since these three metrics are not comparable, we normalized each metric to $[0, 1]$ before calculating the driving performance. Furthermore, we standardized the direction of these metrics, with higher values indicating better performance:

$$\bar{P}_m = [\frac{P_m - \min(P_m)}{\max(P_m) - \min(P_m)}]^n + [1 - \frac{P_m - \min(P_m)}{\max(P_m) - \min(P_m)}]^{1-n}. \tag{4}$$

The boolean indicator $n = \mathbb{I}[m \in \{\text{Efficiency}\}]$ is employed to adjust the direction of driving performance metrics. The ultimate driving performance is derived through the average of normalized metrics for safety, efficiency, and comfort. Pseudocode can be found in the supplementary materials.

**Experimental Design.** We conduct two types of experiments for both planners in the SUMMIT simulator: *Fixed Prediction Ability* and *Fixed Planning Ability*.

1. Fixed Prediction Ability: The planner is required to perform a fixed number of predictions within an interactive simulation environment, regardless of the predictor's execution speed. The objective is to clarify the factor that dominates the disparity between prediction performance and driving performance while ensuring the predictive ability of methods.

Table 3: Trajectory prediction metrics.

| Metric Name | Metric Equation |
|---|---|
| ADE | $\frac{1}{T}\sum_{i=1}^{T}\sqrt{(x_i - \hat{x}_i)^2 + (y_i - \hat{y}_i)^2}$ |
| FDE | $\sqrt{(x_T - \hat{x}_T)^2 + (y_T - \hat{y}_T)^2}$ |
| minADE | $\min_{k \in K} \frac{1}{T}\sum_{i=1}^{T}\sqrt{(x_i - \hat{x}_i)^2 + (y_i - \hat{y}_i)^2}$ |
| minFDE | $\min_{k \in K} \sqrt{(x_T - \hat{x}_T)^2 + (y_T - \hat{y}_T)^2}$ |

2. Fixed Planning Ability: The planner is allocated varying time budgets to simulate a predictor running at different speeds. Three sub-experiments are conducted with tick rates set at 30 Hz, 3 Hz, and 1 Hz. The objective is to investigate the factors, except for prediction accuracy, that represent the predictive ability of methods. This provides an explanation for the remaining disparity between prediction accuracy and driving performance.

We collect 50 scenarios for each predictor in each experiment. For each scenario, we randomly select the start and end point for the ego-agent from one of the four real-world maps provided by the SUMMIT simulator. A reference path of 50 meters is maintained between the two points, and the ego-agent is instructed to follow this path. A certain number of exo-agents including pedestrians, cyclists, and vehicles is randomly distributed within the environment. We implement all selected predictors in the simulator, except for HOME and DSP, due to their significantly longer running time, making the closed-loop evaluation infeasible. These two methods are solely employed in the Sim-Real Alignment, as shown in the supplementary materials. Notably, the RVO planner produces identical results for these two experimental settings since it conducts prediction only once per timestep.

To investigate the correlation between prediction accuracy and driving performance, we train all selected motion prediction models on the Alignment dataset collected from the SUMMIT simulator. We collect 59,944 scenarios and separate them into two groups: 80% training and 20% validation. Each scenario consists of about 300 steps. Subsequently, it is filtered down to 50 steps by taking into account the number of agents and their occurrence frequency. The nearest three agents are randomly selected to be the *interested agent* for prediction. The data collection was conducted on a server equipped with an Intel(R) Xeon(R) Gold 5220 CPU. We use four NVIDIA GeForce RTX 2080 Ti to speed up the running.

## 5 Experimental Results

The experiment part is designed and organized to answer the following questions:*1*) Can the current prediction evaluation system accurately reflect the driving performance?; *2*) What is the main factor causing the disparity between prediction accuracy and driving performance?; and *3*) How do we propose evaluating predictors based on driving performance?

### 5.1 The Limitation of Current Prediction Evaluation

This section illustrates the limitation of current prediction evaluation systems in accurately reflecting realistic driving performance. We use ADE as an example, with FDE results provided in the supplementary materials. We refer to the ADE and minADE calculated in the Alignment dataset as *Static ADE* and *Static minADE* since they are obtained from static evaluation. Fixed prediction ability experiments are conducted for both RVO and DESPOT planners in the SUMMIT simulator.

**Static ADE.** As revealed in Figure 3a, there is no significant correlation between Static ADE and driving performance. Specifically, in the DESPOT planner, we observe a counterintuitive positive relationship. According to this finding, higher Static ADE would imply better driving performance. However, this hypothesis lacks a realistic basis and should be rejected since it exceeds the 95% confidence interval. Static ADE can not serve as a reliable indicator of driving performance.

**Static minADE.** To consider the impact of multi-modal prediction, we further investigate the correlation between Static minADE and driving performance. As depicted in Figure 3b, the multi-modal prediction accuracy better reflects the driving performance. However, in the RVO planner, the linear regression exceeds the 95% confidence interval, thereby weakening the credibility of the evaluation metric. Furthermore, we still observe the positive correlation between prediction error and driving performance in the DESPOT planner, despite the lack of a realistic basis. These two observations prevent us from asserting the efficacy of Static minADE.

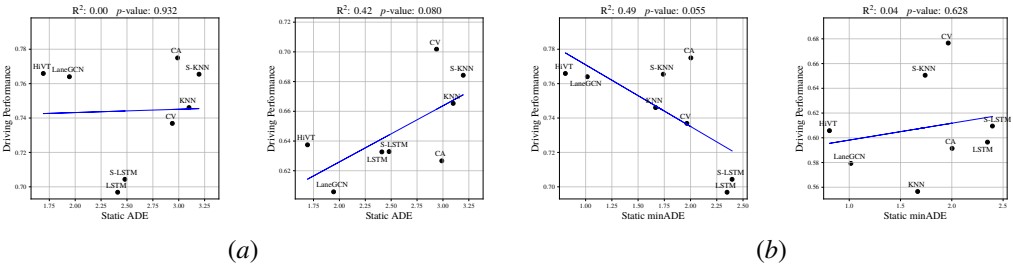

(a)            (b)

Figure 3: The correlation between static prediction metrics and driving performance for RVO (left) and DESPOT (right) planners. (*a*) Static ADE versus driving performance. (*b*) Static minADE versus driving performance. A substantial disparity between prediction accuracy and driving performance is observed, regardless of whether the accuracy is measured by Static ADE or Static minADE.

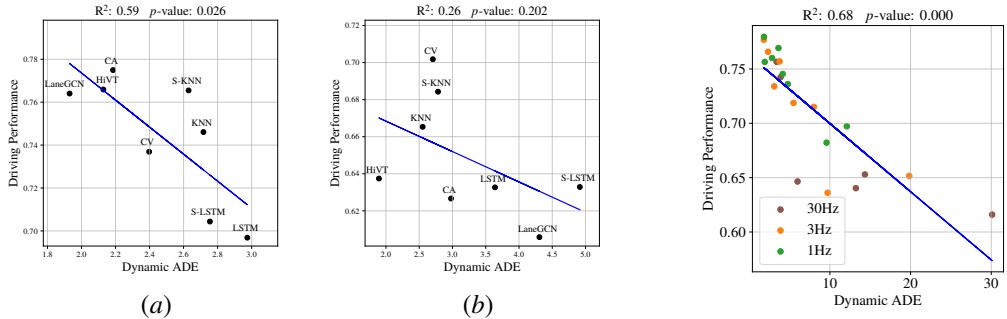

Figure 4: The correlation between Dynamic ADE and Driving Performance with fixed prediction ability. (*a*) In the RVO planner. (*b*) In the DESPOT planner. Due to the mitigation of the dynamics gap, a strong correlation between Dynamic ADE and driving performance is observed for both planners.

Figure 5: The correlation between Dynamic ADE and Driving Performance with fixed planning ability. The correlation becomes weaker when the tick rate is set higher.

In summary, current evaluation metrics merely represent the vanilla accuracy of predictors and do not fully reflect the driving performance. It is urgent to identify the overlooked factor(s) on this issue.

## 5.2 The Dominant Factor: Dynamics Gap

The disparity between static evaluation metrics and driving performance can be attributed to various factors. However, due to the lack of quantitative analysis, it remains unclear which factor primarily contributes to this disparity. In this section, we utilize fixed prediction ability experiments to identify the dominant factor affecting the disparity between prediction accuracy and driving performance. We use *Dynamic ADE* as an example. Similar to Static ADE, Dynamic ADE is calculated as the average L2 distance between the forecasted trajectory and the ground truth. However, the dynamic prediction accuracy in the interactive simulator differs from that of the Alignment dataset, since agents behave differently. This leads to the difference between Dynamic ADE and Static ADE.

**Dynamics Gap.** Figure 4 reports the correlation between Dynamic ADE and driving performance for both RVO and DESPOT planners. Compared to Static ADE, Dynamic ADE displays a significantly stronger correlation with driving performance for both planners. Quantitatively, using the correlation coefficient as a measurement, the dynamics gap accounts for 77.0% of the inconsistency between Static ADE and driving performance in the RVO planner, and 70.3% in the DESPOT planner. This emphasizes the dominant role of the dynamics gap in addressing the challenge of implementing predictors in the real world.

Furthermore, we compare the dynamics gap with factors that influence prediction accuracy and potentially affect driving performance. Except for the multi-modal prediction, the asymmetry of prediction errors [21] and occlusion are selected for comparison. We measure the significance of factors by assessing their impact on the correlation coefficient between prediction accuracy and driving performance. The disparity between static ADE and driving performance is denoted as *Total*.

**Multi-Modal Prediction.** We examine the variation in correlation coefficient from single-trajectory prediction to their multi-trajectory variants.

Table 4: Impacts on the correlation coefficient between prediction accuracy and driving performance.

| Factor | ΔCorrelation Coefficient | |
| --- | --- | --- |
| | RVO | DESPOT |
| Occlusion | -0.22 | 0.16 |
| Prediction Asymmetry | 0.17 | 0.20 |
| Multi-Modal Prediction | 0.70 | 0.45 |
| Dynamics Gap | **0.77** | **1.16** |
| Total | 1.00 | 1.65 |

**Prediction Asymmetry.** We check the change in correlation coefficient from considering all agents' predictions to considering only the interested agent that may affect the ego-plan.

**Occlusion.** We examine the variation in correlation coefficient from considering only agents with complete observations to considering agents with missed observations.

As depicted in Table 4, considering the asymmetry of prediction errors, multi-modal prediction, and dynamics gap can mitigate the disparity between prediction accuracy and driving performance. The consideration of occlusion does not yield satisfactory results in the RVO planner, as the planner mainly focuses on short-term predictions and is minimally affected. The asymmetric of prediction errors has a noticeable impact on both planners, highlighting the importance of predictions involving agents that may affect ego-planning. The multi-modal prediction exerts a significant impact on the RVO planner. However, when a complex planner such as DESPOT is used in conjunction, the impact becomes much weaker. It is reasonable to assume that the sampling-based feature of the DESPOT planner mitigates the influence of prediction uncertainty. Among various metrics, the dynamics gap exhibits superior performance in both planners and effectively resolves the majority of the disparity.

We can conclude that the dynamics gap is the main factor that causes the disparity between prediction accuracy and realistic driving performance. The Dynamic ADE, which is evaluated through interactive simulation environments, is capable of incorporating the dynamics gap and displaying a significant correlation with driving performance.

## 5.3 On the Predictive Ability of Methods

Although the dynamics gap accounts for the majority of the disparity between prediction accuracy and driving performance, the correlation between Dynamic ADE and driving performance is unsatisfactory in the DESPOT planner. To address this issue, we must answer two additional questions: *1*) Does prediction accuracy fully reflect the predictive ability of methods? *2*) If not, what other factor(s) is most important? We examine three well-known but never fully investigated factors and their correlation with driving performance: the temporal consistency of predictions [37], the distribution of prediction errors, and the computational efficiency of predictors.

**Temporal Consistency.** Each prediction scenario contains multiple successive frames within a fixed temporal chunk. Any two overlapping chunks of input data with a small time-shift should produce consistent results. The difference between the predictions of the two overlapping chunks is measured as the temporal consistency, with time-shift set to one frame.

**Error Distribution.** The error distribution of motion prediction contains the mean and variance. While the mean has been captured by the prediction accuracy, we incorporate the variance of prediction errors to account for the remaining impact.

**Computational Efficiency.** The computational efficiency of predictors is determined by their inference time when applied to downstream tasks. Feature extraction time is also included.

According to Figure 6, surprisingly, the temporal consistency of predictions and the distribution of prediction errors show a weak correlation with driving performance in both planners. However, the computational efficiency of predictors exerts a significant influence on the driving performance, as shown in Table 5. In three sub-experiments, the ranking of driving performance is aligned with the ranking of predictors' inference time in the DESPOT planner. We can conclude that the prediction accuracy can not fully represent the predictive ability of methods, and the computational efficiency of predictors is also a significant metric that affects driving performance.

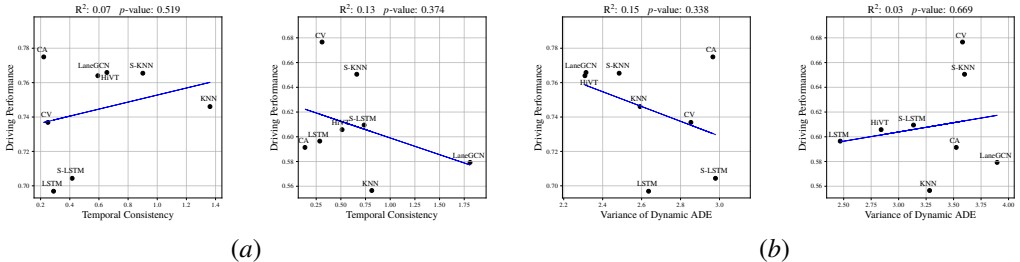

$$(a) \qquad\qquad\qquad\qquad\qquad (b)$$

Figure 6: Contrary to common belief, the correlation between temporal consistency/error distribution and driving performance is not strong for both RVO (left) and DESPOT (right) planners. (*a*) Temporal Consistency versus Driving Performance. (*b*) Error Distribution versus Driving Performance.

Table 5: The correlation between predictors' computation efficiency and driving performance.

| Method | Inference Time ($\downarrow$) | Driving Performance ($\uparrow$) | | |
|---|---|---|---|---|
| | | 30Hz | 3Hz | 1Hz |
| CV | 0.001s | **0.753** | 0.771 | 0.761 |
| CA | 0.001s | 0.739 | **0.774** | **0.779** |
| LSTM | 0.010s | 0.644 | 0.753 | 0.766 |
| S-LSTM | 0.014s | 0.649 | 0.715 | 0.733 |
| HiVT | 0.024s | 0.616 | 0.730 | 0.753 |
| LaneGCN | 0.024s | 0.637 | 0.711 | 0.742 |
| KNN | 0.224s | 0.526 | 0.648 | 0.692 |
| S-KNN | 0.248s | 0.530 | 0.633 | 0.677 |

## 5.4 The Trade-Off between Accuracy and Speed

It should be noted that there is a trade-off between computational efficiency and dynamic prediction accuracy for predictors to derive driving performance. As shown in Figure 5, the correlation between Dynamic ADE and driving performance becomes less strong when the tick rate is set higher. This is indicated by the data points deviating further from the best-fit-line in higher tick rates. At this time, it is the computational efficiency rather than dynamic prediction accuracy that decides the driving performance, as shown in Table 5. When the tick rate is set to 30Hz, the planner cannot generate an optimal solution, whereby the ranking of driving performance is determined by computational efficiency. When the tick rate is set to 3Hz, the CA outperforms CV since they have near-optimal solutions. When the Tick Rate is set to 1Hz, the LSTM also outperforms CV. The driving performance is determined by both dynamic prediction accuracy and computational efficiency of predictors in a trade-off manner, highlighting the significance of task-driven prediction evaluation.

## 6 Conclusion

Our study demonstrates the limitation of current prediction evaluation systems in accurately reflecting realistic driving performance. We identify the dynamics gap as the dominant factor contributing to this disparity. Furthermore, our findings reveal that the ultimate driving performance is determined by the trade-off between prediction accuracy and computational efficiency, rather than solely relying on prediction accuracy. We recommend further research incorporating interactive and task-driven evaluation protocols to assess prediction models. Despite these insights, there is still work to be done in the future. Our study focuses on sampling-based and reactive planners, and incorporating optimization-based or geometric planners could further substantiate our conclusions. Oracle perception data from the simulator can not fully represent the complexities of real-world situations. It would be beneficial to use raw sensor data to understand the comprehensive interplay within the entire AD system.

## Acknowledgments and Disclosure of Funding

This research is supported in part by National Key R&D Program of China under Grant 2020YFB1600202, National Research Foundation of Singapore under its AI Singapore Programme (AISG Award No: AISG2-PhD-2022-01-036[T]), and China Scholarship Council.

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
