# – Supplementary Materials –

## Contents

Table 1: Numerical values for the prediction and driving performances.

| Method | FDE | ADE | Safety | Efficiency | Comfort | Driving Performance |
|--------|-----|-----|--------|------------|---------|---------------------|
| CV [6] | 2.012 | 1.020 | 0.828 | 0.620 | 0.763 | 0.737 |
| CA [6] | 1.984 | 1.005 | 0.870 | 0.652 | 0.802 | 0.775 |
| KNN [3] | 2.565 | 1.291 | 0.847 | 0.606 | 0.785 | 0.746 |
| S-KNN [3] | 2.594 | 1.309 | 0.872 | 0.638 | 0.786 | 0.765 |
| LSTM | 1.706 | 0.859 | 0.851 | 0.605 | 0.634 | 0.697 |
| S-LSTM [1] | 1.905 | 0.963 | 0.844 | 0.630 | 0.638 | 0.704 |
| HiVT [9] | 1.356 | 0.692 | 0.871 | 0.661 | 0.766 | 0.766 |
| LaneGCN [5] | 1.276 | 0.637 | 0.849 | 0.682 | 0.761 | 0.764 |

# A    Details of the Radar Plot

In addition to the radar plot, we present the specific numerical values for the prediction and driving performance metrics to provide a more detailed and comprehensive analysis of the system's performance, as demonstrated in Table 1. The static evaluation metrics, ADE and FDE, are trained and validated on the Alignment dataset collected from the SUMMIT simulator. The task-driven evaluation metrics, including safety, efficiency, comfort, and driving performance, are derived from interactive closed-loop scenarios. The process for calculating these metrics is described in Appendix C.

Results in Table 1 are used to plot the correlation map between ADE/FDE and driving performance, which surprisingly indicates no strong correlation between static evaluation metrics and real driving performance. Moreover, to ensure the comparability between prediction performance metrics and driving performance metrics in the radar plot, we normalize all metrics to the scale of [0, 1]. This facilitates the identification of the performance gap among various predictors and whether it correlates with the current prediction performance metrics.

# B    RVO and DESPOT planners

## B.1    The RVO Planner

The Reciprocal Velocity Obstacle (RVO) planner is developed based on [8], which expands on the concept of velocity obstacles [4] to consider the reactive behaviors of exo-agents. The main idea is to create a set of velocity obstacles for each exo-agent, comprising the range of velocities that would result in a collision. Subsequently, the ego-agent chooses a suitable velocity that is not included in the velocity obstacles. The velocity obstacles for the ego-agent $A$ with respect to an exo-agent $B$ are defined as follows:

$$VO_A(B) = \{\mathbf{v}_A | \lambda(\mathbf{p}_A, (\mathbf{v}_A - \mathbf{v}_B)) \cap (B \oplus - A) \neq \emptyset\} \tag{1}$$

where $\mathbf{p}_A$ is the current position of ego-agent. $\mathbf{v}_A, \mathbf{v}_B$ are the velocities of ego-agent and exo-agent, respectively. $\lambda(\mathbf{p}, \mathbf{v})$ denotes the ray starting at $\mathbf{p}$ and heading in the direction of $\mathbf{v}$. $A \oplus B$ denote the Minkowski sum of $A$ and $B$, and $-A$ denote the ego-agent reflected in its reference point $\mathbf{p}_A$. If $\mathbf{v}_A \in VO_A(B)$, the ego-agent will collide with the exo-agent in time, which means that the chosen $v_A$ should be outside the velocity obstacles of all exo-agents.

Moreover, the RVO is defined as the average of the current velocity of the ego-agent and the velocity in the velocity obstacles:

$$RVO_A(B) = \{0.5 * (\mathbf{v}_A + \mathbf{v}) | \mathbf{v} \in VO_A(B)\} \tag{2}$$

To determine the set of candidate velocities, the planner considers all velocities that lie outside the RVO of all exo-agents. Next, the planner selects the velocity closest to the target velocity among these candidates, which can be formalized as:

$$\mathbf{v}_{new} = argmin_{\mathbf{v} \in \mathbf{V}_{free}} ||\mathbf{v} - \mathbf{v}_{target}||_2 \tag{3}$$

where $\mathbf{V}_{free}$ is the set of candidate velocities, $\mathbf{v}_{target}$ is the target velocity, $||.||_2$ is the Euclidean norm of vectors.

Equation 3 identifies the velocity within a subset of velocities that minimizes the difference with the target velocity, while also ensuring that collisions are avoided. The target velocities of exo-agents are predicted by the motion prediction method $M$. To obtain the target velocity for the ego-agent, we multiply the maximum desired speed $v = 6 \text{ m/s}$ by the unit displacement vector. The direction of the unit displacement vector is determined by subtracting the current frame's position from the position of the next waypoint on the reference path.

### B.2   DESPOT Planner

The Determinized Sparse Partially Observable Tree [7] (DESPOT) is a planner for online POMDP (Partially Observable Markov Decision Process) planning. It utilizes a sparse approximation of the standard belief tree to facilitate anytime online planning under uncertainty. The algorithm overcomes two main challenges of POMDP planning: the "curse of dimensionality" (i.e. large state space) through sampling, and the "curse of history" (i.e. long planning horizon) through anytime heuristic search. In the subsequent section, we provide an overview of the DESPOT planner.

*1) State and Observation*: The states in the DESPOT planner contain both continuous-domain physical states and discrete-domain hidden states.

- Continuous states of the ego-agent: $s_A = (x_A, y_A, v_A, \theta_A)$, in which $(x_A, y_A)$ represent the position, $v_A$ is the velocity, $\theta_A$ is the heading direction of the ego-agent.
- Continuous states of exo-agents: $s_i = (x_i, y_i, v_i, \theta_i)$, which includes the position $(x_i, y_i)$, the current velocity $v_i$, and the heading direction $\theta_i$ for the exo-agent $i$ in the set of agents $i \in \{1, \ldots, n\}$.
- Hidden states of exo-agents: $h_i = (t_i, \mu_i)$, $t_i$ is the intention and $\mu_i$ is the intended path of the driver of the exo-agent $i$. These two values can only be inferred from history.

*2) Action*: The action space for DESPOT contains three accelerations: {*Accelerate*, *Decelerate*, *Maintain*}, in which the values for the first two accelerations are $3 \text{ m/s}^2$ and $-3 \text{ m/s}^2$. The maximum speed of the ego-vehicle is $6 \text{ m/s}$.

*3) Transition Function*: Given the current state of the system $s = (s_A, s_1, \ldots, s_n, C)$ and the action of the ego vehicle $a \in \mathcal{A}$, the planner uses the motion predictor $\hat{s}_i = M(s_i, C^t)$ to predict the next state $\hat{s}_i$ for each exo-agent $i$. For the ego-agent, we adopt the bicycle model, a kinematic model with two degrees of freedom, to get the next state $\hat{s}_A$. Additionally, Gaussian noise is introduced to the displacement of each agent to model the stochasticity of human behaviors.

*4) Reward Function*: The reward in DESPOT handles safety, efficiency, and comfort metrics. The details can be found in [2]. For safety, we assign a huge penalty $R_{col} = -1000 \times (v^2 + 0.5)$ when the vehicle collides which is quadratically increased with the collision speed. For efficiency, we assign a speed penalty $R_v = 4(v - v_{max})/v_{max}$ to encourage driving at maximum speed. For comfort, we impose a smoothness penalty $R_{acc} = -0.1$ for each deceleration to penalize jerks, and a penalty of $R_{change} = -4$ for lane changes.

The DESPOT planner can efficiently handle large state spaces and partial observability which makes it an ideal algorithm for planning in complex and dynamic environments. They can also handle state and action uncertainty associated with real-world driving scenarios, such as traffic congestion, unexpected obstacles, and changes in road conditions. By maintaining a belief state and updating it at each timestep, DESPOT can make informed decisions even in situations where the environment is uncertain or partially observable. Finally, DESPOT's reward function captures safety, comfort, and efficiency aspects, making it suitable for real-time decision-making.

## C   Pseudocode for Calculating the Driving Performance

**Safety.** At each time step, the pseudocode in Algorithm 1 checks for collisions between the ego-agent and exo-agents. The states of the ego-agent and exo-agents are contained in dictionaries, including their *x* and *y* positions, as well as the *width* and *length* bounding box sizes and *heading*. To account for the low likelihood of collisions caused by the DESPOT planner, a *buffer* variable is added to

---

**Algorithm 1:** Pseudocode for calculating the collision rate

---

**Input:** agent: dict, buffer: float
**Output:** rotated_corners
1: **Function** GetCorners(*agent, buffer*):
2:   width, length ← agent['bb']
3:   x, y ← agent['pos']
4:   heading ← agent['heading']
5:   dx ← length / 2
6:   dy ← width / 2
7:   corners ← [[x - dx - buffer, y - dy], [x + dx + buffer, y - dy], [x + dx + buffer, y + dy], [x - dx - buffer, y + dy]]
8:   rotated_corners ← empty list
9:   **for** *corner* ∈ *corners* **do**
10:    x_diff ← corner[0] - x
11:    y_diff ← corner[1] - y
12:    new_x ← x + (x_diff * cos(heading) + y_diff * sin(heading))
13:    new_y ← y + (x_diff * sin(heading) - y_diff * cos(heading))
14:    append [new_x, new_y] to rotated_corners
15:   **return** *rotated_corners*

**Input:** ego: dict, exo: dict, buffer: float
**Output:** collision status
16: **Function** CheckCollision(*ego, exo, buffer=1*):
17:   ego_corners ← GetCorners(*ego, buffer*)
18:   exo_corners ← GetCorners(*exo, buffer*)
19:   ego_polygon ← Polygon(ego_corners)
20:   exo_polygon ← Polygon(exo_corners)
21:   **return** *ego_polygon.intersects(exo_polygon)*

---

---

**Algorithm 2:** Pseudocode for calculating the average speed.

---

**Input:** ego_dict: dict
**Output:** average_speed
1: **Function** AverageSpeed(*ego_dict*):
2:   speeds ← [ego_data['speed'] for timestep, ego_data ∈ ego_dict.items()]
3:   average_speed ← np.mean(np.abs(speeds))
4:   **return** *average_speed*

---

---

**Algorithm 3:** Pseudocode for calculating the jerk.

---

**Input:** accel_data, delta_time
**Output:** jerk
1: **Function** CalculateJerk(*accel_data, delta_time*):
2:   jerk ← np.diff(accel_data, axis=0) / delta_time
3:   jerk ← np.linalg.norm(jerk, axis=1)
4:   **return** *jerk*

---

the bounding boxes of all agents to check collisions. This buffer is only incorporated in the length direction, focusing only on the heading. The collision rate is computed by dividing the number of collisions by the total timesteps. The safety is represented by the normalized collision rate.

**Efficiency.** The pseudocode for calculating the average speed is presented in Algorithm 2, which is a critical efficiency metric for autonomous vehicles. In our experiments, the planner controls the speed of the ego-agent while the pure-pursuit algorithm adjusts the steering angle. The speed directly influences the ego-agent's ability to navigate traffic, maintain a safe distance from other vehicles, and promptly reach its destination. A higher speed is usually desirable as it permits vehicles to attain their objectives quickly. The efficiency is represented by the normalized average speed.

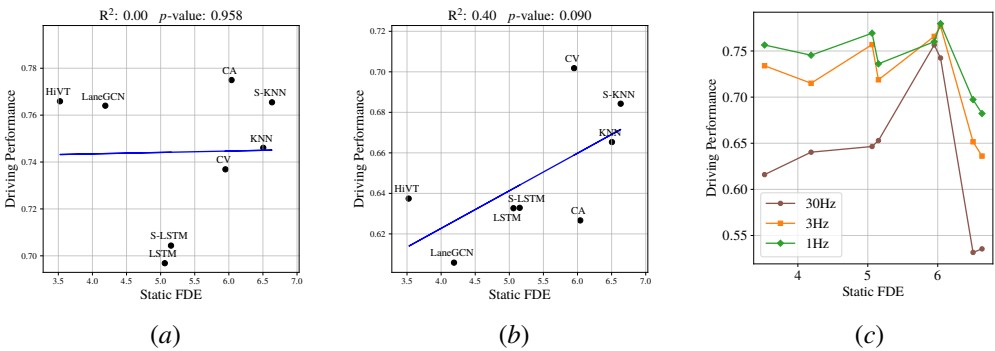

(a)            (b)            (c)

Figure 1: Relationship between Static FDE and Driving Performance. (*a*) Fixed number of predictions/Fixed planning time for the RVO planner: the planner calls predictors once in an action, whereby these two experiments result in the same output. (*b*) Fixed number of predictions for the DESPOT planner. (*c*) Fixed planning time for the DESPOT planner. In total, we found no strong correlation between driving performance and Static FDE.

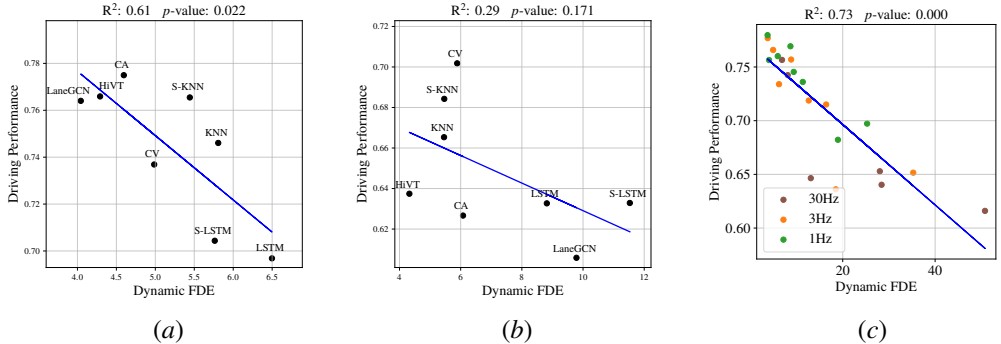

(a)            (b)            (c)

Figure 2: Relationship between Dynamic FDE and Driving Performance. (*a*) Fixed number of predictions/Fixed planning time for the RVO planner. (*b*) Fixed number of predictions for the DESPOT planner. (*c*) Fixed planning time for the DESPOT planner. A much stronger correlation between Dynamic FDE and driving performance is shown for both RVO and DESPOT planners, which can be attributed to the inclusion of dynamics gap in (*a*), (*b*), as well as computational efficiency in (*c*). The correlation is weaker when the planning time budget is tight.

**Comfort.** The pseudocode for calculating jerk, which measures the rate of change of acceleration, is presented in Algorithm 3. Jerk is particularly sensitive to abrupt changes in motion and is important in capturing passengers' discomfort caused by sudden accelerations or decelerations. It enables a clear differentiation between smooth and rough motions and offers a more precise understanding of comfort compared to acceleration or velocity measures. The jerk is first averaged across all timesteps and then normalized to represent comfort.

**Normalization.** To ensure comparability across safety, efficiency, and comfort metrics, we apply a simple normalization technique that scales each metric to a range of [0, 1]. To achieve this, we subtract the minimum value from each metric and divide the result by the range. Additionally, we normalize the direction of these three metrics, where higher values represent better performance. The process is as follows:

$$\bar{P}_{\text{metrics}} = \begin{cases} \frac{P_{\text{metrics}} - P_{\text{metrics}}^{\min}}{P_{\text{metrics}}^{\max} - P_{\text{metrics}}^{\min}}, & \text{metrics} = \{\text{efficiency}\} \\ 1 - \frac{P_{\text{metrics}} - P_{\text{metrics}}^{\min}}{P_{\text{metrics}}^{\max} - P_{\text{metrics}}^{\min}}, & \text{metrics} = \{\text{safety}, \text{comfort}\} \end{cases} \tag{4}$$

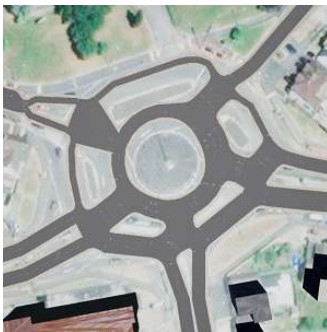

(a) Magic (UK) Roundabout

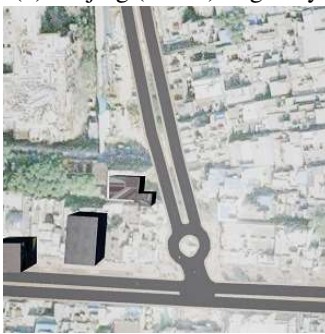

(b) Beijing (China) Highway

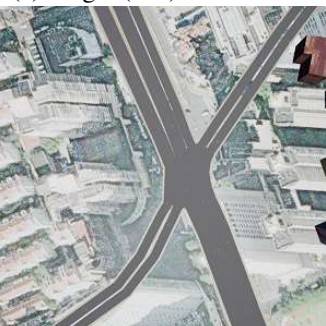

(c) Shi-Men-Er-Lu (China) Intersection

(d) Chandni-Chowk (India) Roundabout

Figure 3: Four maps provided by the SUMMIT simulator. Each map encodes different complexities, including roundabout, highway, and intersection.

Table 2: Impacts on the correlation coefficient between prediction accuracy and driving performance.

| Factor | ΔCorrelation Coefficient | |
|---|---|---|
| | RVO | DESPOT |
| Occlusion | -0.23 | 0.06 |
| Prediction Asymmetry | 0.14 | 0.08 |
| Multi-Modal Prediction | 0.73 | 0.49 |
| Dynamics Gap | **0.78** | **1.17** |
| Total | 1.00 | 1.63 |

where $P_.^{\min}$ and $P_.^{\max}$ represent the minimum and maximum pairs of each performance metric among all scenarios.

**Driving Performance.** The driving performance is obtained by averaging the normalized safety, efficiency, and comfort.

## D   Maps

We use the maps introduced in Figure 3 to collect data for the Alignment dataset and conduct reactivate experiments using both planners on the SUMMIT simulator. The maps are obtained from different cities worldwide with varied complexities such as intersections, roundabouts, and highways. We carried out experiments on a server with an Intel(R) Xeon(R) Gold 5220 CPU, which has 36 physical cores and 72 threads, and four NVIDIA GeForce RTX 2080 Ti GPUs.

## E   Experiment Results for FDE

This section presents the experimental results for FDE in terms of driving performance, following the experimental setup and result format similar to ADE. As demonstrated in Figure 1, Static FDE is not

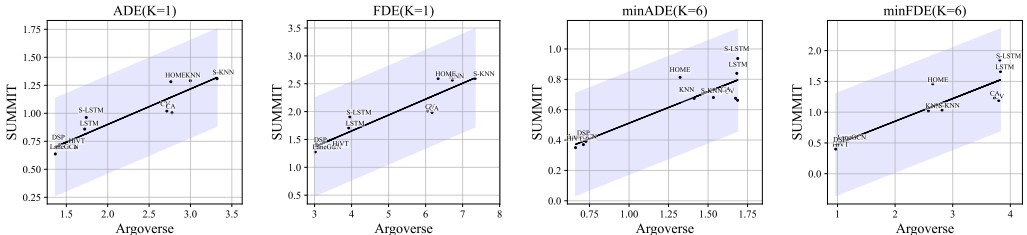

Figure 4: The prediction performance of all selected prediction methods are aligned between Argoverse and Alignment datasets. All data points fall within the 95% confidence interval and conform well to linear regression.

strongly related to driving performance, whereas dynamic FDE shows a much stronger correlation in both experiments. This can be attributed to the ignorance of the dynamics gap between the dataset and real driving scenario and the computational efficiency of predictors, as shown in Figure 2. The trade-off between computational efficiency and dynamic prediction accuracy still exists in the experiments of FDE. As shown in Figure 2c, the correlation between Dynamic FDE and driving performance becomes less strong when the tick rate is set higher. The computational efficiency of predictors should also be considered when the time budget is tight.

The potential factors that could impact Dynamic FDE and their correlation coefficients to driving performance are presented in Table 2. It is recommended to evaluate prediction models using Dynamic FDE with the closest agents in the interactive closed-loop scenarios.

## F  Sim-Real Alignment

To demonstrate the alignment between the SUMMIT simulator and the real world, we train and evaluate all selected motion prediction models on both the Argoverse dataset [3] and the Alignment dataset collected from the SUMMIT simulator. We collect 59,944 scenarios and separate them into two groups: 80% training and 20% validation. Each scenario consists of about 300 steps. Subsequently, it is filtered down to 50 steps by taking into account the number of agents and their occurrence frequency. The nearest three agents are randomly selected to be the *interested agent* for prediction.

Figure 4 illustrates the comparison of prediction performance between the Argoverse and Alignment datasets. The R-squared values of the four subplots are 0.798, 0.777, 0.855, and 0.844, respectively. These values indicate that the majority of variation can be explained by the linear relationship between the prediction performance in these two datasets. Furthermore, the p-values are all less than 0.01, providing strong support for the statistical significance of the alignment. The consistent results suggest that the Argoverse and Alignment datasets share similar underlying features. Therefore, the SUMMIT simulator can be employed to evaluate real-world prediction performances. Likewise, the alignment of driving performance is verified via the simulator itself. Thus, we take effort in identifying the optimal simulator, SUMMIT, which is built upon Carla, the most widely-used simulator in recent competitions and research.