# OpenReview forum: "What Truly Matters in Trajectory Prediction for Autonomous Driving?"
_NeurIPS.cc/2023/Conference — NeurIPS 2023 poster_

### Official Review · Reviewer_1Qcn · 2023-07-06

**Soundness:** 3 good
**Presentation:** 3 good
**Contribution:** 4 excellent
**Rating:** 7
**Confidence:** 4

**Summary:**

This paper presents a timely and well-executed study on motion prediction, focusing on the relationship between the dominant evaluation paradigm (static, offline metrics) and the true objective of research in this domain (safer planning). The study uncovers several important and surprising findings: (1) static offline metrics are not correlated to planning performance, (2) evaluating on the frames observed by the planner (dynamic offline metrics) are significantly better correlated, (3) computational efficiency in prediction is increasingly important for sophisticated planners, and (4) simple baselines for prediction (constant velocity/constant acceleration) perform best when considering downstream planning performance.

**Strengths:**

The study setup is excellent, making well-motivated choices for the task, simulator, predictors, planners, metrics, etc. The writing is clear. The presentation of the key findings both visually and numerically also makes this paper an interesting and easy-to-understand read. Most importantly, it highlights a flaw in the widely used metrics that the community optimizes motion prediction models for, and shows that naive baselines are sufficient to outperform SoTA learned forecasting when using the right metrics.

**Weaknesses:**

Overall, the paper has very few weaknesses in my opinion. While the selected ML prediction models do not include the current leaderboard winners, I believe they are representative. The experiments could have been conducted on more realistic scenarios (e.g. the nuPlan simulator), but Section 5.1 shows that the simulator used is sufficiently aligned to real datasets for the purpose of this study. However, some technical details regarding the study are unclear (please see the “Questions” section #2, #3, #4 for specifics).

**Questions:**

1. A very interesting finding that I suggest should be highlighted more is that in all closed-loop evaluation settings, the best among the prediction methods used in this study is always either CV or CA - the naive baselines.
2. Do the CV and CA baselines forecast the positions along a straight line (heading direction), or do they take into account angular speed/acceleration as well?
3. In L316, could you please elaborate on what "complete/incomplete observations" refers to?
4. L328 recommends Dynamic FDE with Closest, when closed loop simulation is possible. However, why would this be preferable to simply ranking prediction methods by the “Driving Performance” of the planner?
5. Please cite https://arxiv.org/abs/1809.04843 while wrapping up Section 2.2 - this is a well-known study with a similar setup, aimed at planning instead of motion prediction. It would also be interesting to contrast your findings to theirs, given the change in task.
6. Could part of the discrepancy arise from the fact that prediction models penalize errors on all vehicle instances equally, while some are less relevant for planning than others? The result in Table 4 showing the improved correlation when considering only the nearest agents seems to indicate this.
    1. Could you add a row with the “Closest” checkmark for the static, multimodal ADE?
    2. If it is possible to include a task-aware offline metric such as PKL [15], CAPO [18] or some equivalent in the analysis, this would add significant value to the paper, and I would be happy to increase my rating.

Minor:

1. L078: Lastname et al. instead of Firstname et al.

Update:

Thank you for taking the time to carefully answer my questions, I appreciate the effort. The rebuttal clarifies most of my questions. I understand the choice of Dynamic ADE/FDE for the experiments in this paper (as the rebuttal points out, "a prediction metric that employs the same calculation methodology but behaves distinctively in these two evaluation frameworks" helps provide evidence for their claims).

I am still leaning towards a positive rating as I see the main contribution of this paper as highly relevant to the motion prediction community for driving. This is a large research area with hundreds of published papers, and little incentive to move away from static metrics (even though the lack of alignment between open-loop and closed-loop metrics for planning is reasonably well-known). This paper opens up a discussion on the inadequacy of the prevalent motion forecasting metrics, and could be a valuable first step towards more comprehensive evaluation of prediction models in the future.

**Limitations:**

Limitations are discussed in Section 6.

---

> ### Author Rebuttal · Authors · 2023-08-09
>
> We thank the reviewer for the comprehensive comments! We kindly ask the reviewer to let us know if further clarification or information is needed.
>
> >A very interesting finding that I suggest should be highlighted more is that in all closed-loop evaluation settings, the best among the prediction methods used in this study is always either CV or CA - the naive baselines.
>
> Yes, that’s an important observation! When the time budget is limited, the simple rule-based model gives very good driving performance, which is in line with previous research: a simple Constant Velocity Model can outperform even state-of-the-art neural models [1]. We will include this paper as a reference in our final version.
>
> >Do the CV and CA baselines forecast the positions along a straight line, or do they take into account angular speed/acceleration as well?
>
> That’s a good question. We solely utilize position for our predictions, extracting speed and acceleration information. As a result, angular speed or acceleration is not taken into consideration. This approach aligns with the baseline of the Argoverse Competition.
>
> >In L316, could you please elaborate on what "complete/incomplete observations" refers to?
>
> The term "complete" signifies the presence of all 20 observations of an agent, while "incomplete" denotes instances where certain observations are absent due to factors like occlusion, being outside the receptive field, etc.
>
> >L328 recommends Dynamic FDE with Closest, when closed loop simulation is possible. However, why would this be preferable to simply ranking prediction methods by the “Driving Performance” of the planner?
>
> This is a great point for us to clarify more! The correlation between Dynamic ADE/FDE and driving performance is strong for both planners, rendering them suitable for ranking prediction models. However, the core contribution of this paper lies in revealing the absence of correlation between static evaluation and driving performance. To illustrate this, we adopt a prediction metric that employs the same calculation methodology but behaves distinctively in these two evaluation frameworks. By showing this, we emphasize the significance of dynamic evaluation.
>
> >Please cite https://arxiv.org/abs/1809.04843 while wrapping up Section 2.2 - this is a well-known study with a similar setup, aimed at planning instead of motion prediction. It would also be interesting to contrast your findings to theirs, given the change in task.
>
> Thank you for bringing this paper to our attention. We concur with the significant factors highlighted within it, like the asymmetry of prediction error. We will include this paper in related works and contract our findings with it in Q6.
>
> >Could part of the discrepancy arise from the fact that prediction models penalize errors on all vehicle instances equally, while some are less relevant for planning than others? The result in Table 4 showing the improved correlation when considering only the nearest agents seems to indicate this.
>
> Yes. Even when two prediction models present the same ADE, their driving performances are not identical. This discrepancy arises from the different influence of prediction errors to planning. For instance, errors associated with the closest exo-agent can have a higher impact on the ego-agent's plan. The conflict between the predicted state of the exo-agent with the ego-agent's plan also matters.
>
> **However, it is important to note that the asymmetry of prediction error is not universal, which means it may occur only in specific scenarios or for specific agents**. As a result, the influence of this factor is less significant than dynamic evaluation and computational efficiency, both of which impact prediction evaluation are ubiquitous. To provide evidence, we have accounted for the most critical aspect of asymmetry prediction error by comparing "Closest Dynamic ADE" with "Dynamic ADE." Consequently, the influence of asymmetry on the correlation between ADE and Driving Performance is less than dynamic evaluation. (Influence For RVO Planner: 0.16 vs 0.61; Influence For DESPOT Planner: 0.20 vs 0.96)
>
> Hence, in this paper, our primary focus is on discussing the significance of dynamic evaluation and computational efficiency while giving less emphasis to this particular factor, even though it remains crucial in prediction evaluation.
>
> >Could you add a row with the “Closest” checkmark for the static, multimodal ADE?
>
> The static and multimodal ADEs are already “Closest”. Within the Alignment dataset, we randomly select the three nearest agents as the “interested agent” for prediction, with predictions solely made for this "interested agent" in each scenario. This approach is implemented to mirror the configuration of an interested agent as in the Argoverse dataset.
>
> >If it is possible to include a task-aware offline metric such as PKL [15], CAPO [18] or some equivalent in the analysis, this would add significant value to the paper, and I would be happy to increase my rating.
>
> These task-aware offline metrics offer valuable insights in open-loop evaluation where static datasets are available. However, applying these metrics poses challenges within closed-loop, where the ego planner functions in real-time, causing the ground truth to dynamically shift based on the planner's choices. This dynamic nature eliminates the existence of a fixed ground truth, against which to benchmark the planner's performance. Given this absence of a consistent ground truth within closed-loop, conducting PKL or CAPO measurements becomes complicated. While we recognize the value of task-aware offline metrics, we will explore means of adapting such measures for closed-loop evaluation in future.
>
> >L078: Lastname et al. instead of Firstname et al.
>
> Thank you for bringing this error to our attention. We will correct this to "McAllister et al." in our final version.
>
> [1] Schöller et al. What the constant velocity model can teach us about pedestrian motion prediction. RAL 2020.

---

> > ### Comment · Reviewer_1Qcn · 2023-08-19
> >
> > Thank you for taking the time to carefully answer my questions, I appreciate the effort. The rebuttal clarifies most of my questions. I understand the choice of Dynamic ADE/FDE for the experiments in this paper (as the rebuttal points out, "a prediction metric that employs the same calculation methodology but behaves distinctively in these two evaluation frameworks" helps provide evidence for their claims). However, I am still missing an answer as to whether the Dynamic ADE/FDE has any benefit as a metric to a practitioner building prediction systems for driving scenes. The same question has been raised by reviewer Aygk, and I am curious to see the answer.
> >
> > I am still leaning towards a positive rating as I see the main contribution of this paper as highly relevant to the motion prediction community for driving. This is a large research area with hundreds of published papers, and little incentive to move away from static metrics (even though the lack of alignment between open-loop and closed-loop metrics for planning is reasonably well-known). This paper opens up a discussion on the inadequacy of the prevalent motion forecasting metrics, and could be a valuable first step towards more comprehensive evaluation of prediction models in the future.

---

> > > ### Author Response · Authors · 2023-08-20
> > >
> > > Thanks for your recognition of our work. In this paper, our main contribution stands on:
> > >
> > > 1. Establish the existence and significance of the dynamics gap and computational efficiency.
> > >
> > > 2. Emphasize the efficacy of the alternative evaluation protocol (simulation) when real-world tests are unaffordable.
> > >
> > > In this perspective, dynamic ADE/FDE serve solely as the tools we employ to substantiate these two contributions, rather than constituting our primary contribution. The difference in the correlation coefficient between dynamic ADE/FDE and static ADE/FDE versus driving performance is notable, demonstrating the significance of the dynamics gap. Similarly, the importance of computational efficiency is emphasized by the remaining gap between dynamic ADE and driving performance.
> > >
> > > However, the benefit of dynamic ADE to a practitioner building autonomous driving systems is also significant, as the primary step in solving dynamic gap is to perceive it. Dynamic ADE versus static ADE can serve as a metric to value how much the dynamics gap is solved. We present a potential application here: motion model evaluation.
> > >
> > > Motion Model Evaluation: While each simulator employs its unique motion model for exo-agents, the assessment of motion model quality remains challenging. Through the utilization of dynamic ADE, three values can be computed utilizing the same predictor and planner: static ADE within the simulation dataset, dynamic ADE within the simulation environment, and dynamic ADE during real-world testing. These metrics solely involve changes in the motion model of exo-agents, enabling the evaluation of the simulation's fidelity to real-world agents' motion. e.g.
> > >
> > > $$
> > > fidelity =\frac{ADE_{real} - ADE_{sim}}{ADE_{real} - ADE_{static}}
> > > $$
> > >
> > > We would like to thank the reviewer for thoughtful and positive comments. We would be more than glad to discuss any remaining concerns you might have.

---

> > > > ### Comment · Reviewer_1Qcn · 2023-08-21
> > > > **Follow-up question**
> > > >
> > > > Thanks again for the clarification. I concur on the significance of the 2 main contribution points, and I think this paper provides evidence to the community that evaluating prediction models through simulation is worth the effort. While the motion model evaluation idea is interesting, it remains secondary to the main contributions, so I would suggest mentioning it in the supplementary material.
> > > >
> > > > I have one final follow-up question: will the code to reproduce your experiments be made available? If so, am I correct in understanding that with access to this code, future work on motion prediction could plug in their predictors to this code and be able to report Dynamic ADE on SUMMIT?
> > > >
> > > > Given the mixed reviews, I believe that the paper could have done a better job at emphasizing the significance of the main contributions. Nevertheless, I think the discussion phase will help the authors better present this point, and I lean towards maintaining my initial rating of "Accept".

---

> > > > > ### Author Response · Authors · 2023-08-21
> > > > >
> > > > > Thanks for your active engagement in the discussion. We deeply appreciate your thorough review of all the comments,  as well as the valuable insights and recognition regarding our core contributions. We fully concur that highlighting the dominant influence of dynamics gap and computational efficiency on driving performance is crucial in our main text, as it distinguishes our work from previous studies within the field. The motion model evaluation idea and Sim-Real Alignment experiment should be moved to the supplementary material.
> > > > >
> > > > > For the follow-up question, code release is currently in progress. We have tidied up the code for predictor implementation and scenario analysis. Given the complexity for simulation with planners, we have opted to leverage Docker to facilitate future application of our work. This may take some additional time. We aim to publish our code soon to provide an interactive evaluation protocol for prediction methods.

---

### Official Review · Reviewer_RCyQ · 2023-07-07

**Soundness:** 3 good
**Presentation:** 3 good
**Contribution:** 2 fair
**Rating:** 4
**Confidence:** 3

**Summary:**

The authors examine the discrepancy between trajectory prediction accuracy and driving performance in the task of autonomous driving when a fixed dataset is used and in the presence of multiple surrounding traffic participants (such as other vehicles). Specifically, they assert that this discrepancy arises from:
-  A dynamics gap between the static dataset that the trajectory predictor is trained on and its subsequent usage for planning in real-world driving scenario. This is because the ego-vehicle’s planner may act differently to the static dataset, which results in novel reactions from the other entities in the environment (vehicles, pedestrians, etc.) and thus out-of-distribution scenarios occur.
- Computational efficiency issues caused by the planner and trajectory predictor being too slow to accommodate real-time operation.

Although the work does not propose a novel method, the authors conduct of study of existing methods to illustrate the importance of a dynamic approach where the ego-vehicle collects data with the specific predictor (instead of relying on a single fixed static dataset) and efficient trajectory prediction.

A wide variety of trajectory prediction models are used (Table 1) to ensure high test coverage along with two planners (RVO and DESPOT) (section 4.2)

For trajectory prediction evaluation they use the Average/Final Distance Error (ADE/FDE) in Table 2. To evaluate driving performance a combination of collision avoidance, speed to goal and comfort (minimize jerk) are considered (section 4.4).

Experiments first attempt to illustrate how their chosen simulator (SUMMIT, extended from Carla) is suitable to evaluate real-world performance based on simulation (section 5.1, Figure 2) as trajectory prediction simulation results (SUMMIT) and real-world dataset results (Argoverse dataset) are positively correlated.

They then illustrate the performance gap that occurs when evaluating against a static dataset versus one collected interactively (dynamically) from simulation where the ego vehicle operates with the predictor (section 5.2, Figure 3. versus Figure 4). Most notably, they find that efficiency concerns are most prevalent under a high planner-simulation tick rate. In this case the predictor efficiency has the greatest effect on driving performance instead of the dynamics gap since it cannot keep up with the high tick rate (Table 3). Table 4 further shows that the discrepancy between driving performance and trajectory prediction accuracy (ADE) is reduced when only nearby agents with full observations is preferred in addition to dynamic data collection.

**Strengths:**

- Experiments are well done and compare a variety of approaches (ten trajectory prediction models, two planners).

- Claims are backed clearly by data. The results show the value of using an interactive simulation environment for data gathering and favoring efficient predictors when doing real-time planning for autonomous driving.

- In general, well written and clear.

**Weaknesses:**

- The work does not propose a novel approach but is instead of survey of existing methods.

- It could also be argued that the survey results are somewhat obvious (which compounds with the first criticism): 1) relying solely on a static dataset for model-based training may result in a distributional shift during real-world evaluation once planning is done, causing failure and 2) efficient predictors are important for real-time planning.

Minor Criticism:

- The combination of performance metrics $P_{\mathrm{Safety}}, P_{\mathrm{Efficiency}}, P_{\mathrm{Comfort}}$ could be clarified in the text (section 4.4). Sensibly, $P_{\mathrm{Safety}}$ and $P_{\mathrm{Comfort}}$ are to be minimized while $P_{\mathrm{Efficiency}}$ is maximized. But all metrics are labeled “performance” so I was originally expecting them to be handled uniformly. The text does mention that “we normalized the direction of these three metrics” but this is still somewhat ambiguous. Things are fully clarified once looking at the supplemental material where the sign on $P_{\mathrm{Safety}}$ and $P_{\mathrm{Comfort}}$ are reversed during normalization. However, I would find it helpful if this was more clearly mentioned to in the text.

- I feel that the first experimental section (section 5.1) is perhaps not relevant to the main purpose of the paper besides illustrating that trajectory prediction accuracy in simulation and the real-world are positively correlated across different models. I imagine most semi-photorealistic simulation environments would possess this property, especially given its extension from the mature Carla simulator. They also do not explicitly show that driving performance is ultimately related (only trajectory prediction performance). There does appear to be a notable ADE/FDE error increase when moving to the real-world dataset. Thus, once applied to planning, I could see a case where a model might perform “well” in simulation but “poorly” in the real world (despite performing better than its peers).

**Questions:**

- Should the minADE and minFDE equations in Table 2 be a function of k? I assume that each predicted  $\hat{x}_i$ is a sample taken K times?

- In Table 3, is the “Inference Time” for one network inference loop and not the overall time for planning? If it is the overall time for planning I would be confused as to why performance changes for some models below the 30hz refresh rate when going from 30 to 3 to 1 hz.

- In section 5.1, lines 237-238 it says that the “nearest three agents are randomly selected to be the interested agent for prediction”. Is there a reason why only 3 agents were considered? Was this the best trade-off between computation time and performance?

**Limitations:**

The authors did not explicitly list any limitations of their method. However, their work is primarily a study of existing methods and so a limitations section is perhaps less relevant. Nonetheless, one possible limitation is the focus solely on simulation for illustrating static versus dynamic driving performance discrepancy. Please refer to the weaknesses section for more possible limitations.

---

> ### Author Rebuttal · Authors · 2023-08-09
>
> We thank the reviewer for the valuable comments! We kindly ask the reviewer to let us know if further clarification or information is needed.
>
> >The work does not propose a novel approach but is instead a survey of existing methods.
>
> We do not provide a new approach, but instead a prediction evaluation protocol, which is also important in the field of autonomous driving. Proper evaluation is an important yet often overlooked aspect of machine learning research. Prior works focusing on evaluation from NeurIPS [1,2] play an important role in their respective field.
>
> >It could also be argued that the survey results are somewhat obvious.
>
> This is a great point for us to elaborate more! This paper shed light on **the equally essential need for evaluating seemingly static modules, such as prediction, through closed-loop evaluation**, which has been overlooked in recent studies [3,4] and SOTA of Argoverse Competition [5]. Another important idea is **the trade-off between computational efficiency and accuracy even when the predictor is fast enough for the planner's execution**. In the context of autonomous driving, a consensus exists that the perception system must meet a specified threshold for proper planning. Once this threshold is satisfied, the focus shifts to ensuring accuracy. (e.g., 100ms [6,7]) However, our experiments demonstrate the trade-off persists even below the threshold. When the predictor runs much faster than the threshold (2ms), the driving performance remains dominated by the fastest prediction method (CV). This trade-off holds importance, as various planners exhibit their trade-offs in addition to their fundamental requirement.
>
> >The combination of performance metrics could be clarified in the main text.
>
> Sure! We will put the explanations in our main text to make it clearer, detalis in the general response due to word limitation.
>
> >The first experimental section is not relevant to the main purpose of the paper.
>
> As pointed out, our first experiment is to establish the positive correlation between trajectory prediction accuracy across various environments. This serves as the foundation of our experiments, but not supporting the main idea. **To avoid misunderstanding of our core concept, we will include the Sim-Real Alignment experiment in the Appendix.**
>
> >A model might perform “well” in simulation but “poorly” in the real world.
>
> The gap you mentioned arises from the disparities between simulation and real-world, which we can refer to as interaction gap. This mismatch arises due to unreal exo-agent behavior, as the simulator controls their actions. Unfortunately, this can only be addressed through real-world tests, which is often unaffordable. Our paper focuses on addressing the dynamics gap that exists between static evaluation and dynamic evaluation. In static evaluation, both interaction gap and dynamics gap remain unresolved. In dynamic evaluation, only the interaction gap remains. This is our main contribution, and how to solve the interaction gap lies beyond the scope of this paper.
>
> >Should the minADE and minFDE equations in Table 2 be a function of k? I assume that each predicted  xi Is a sample taken K times?
>
> Thank you for bringing this to our attention. We apologize for the error in formulating minADE/FDE. We take our experiment setting based on the consensus of Argoverse/Waymo Dataset, setting K=6. We will explicitly state it in the final version.
>
> >Is the “Inference Time” for one network inference loop and not the overall time for planning?
>
> Yes. The inference time is for one network inference loop and not the overall time for planning. It is essential to distinguish the tick rate as the time of simulator, not the inference time. In a simulator with Tick Rate=30Hz, planners are constrained to act within 0.3s (real time). Conversely, Tick Rate=3Hz allows 3s for planning. Adjusting the tick rate grants planners more time to act, enabling them to explore more potential future states, thereby influencing driving performance.
>
> >Is there a reason why only 3 agents were considered? Was this the best trade-off between computation time and performance?
>
> This setting is designed to replicate the data generation process of Argoverse. In the Argoverse dataset, each scenario selects one agent as the interested agent, which must possess complete observation and future states. As the nearest agents are more likely to have full observation, we choose it as the interested agent in our data generation. To preserve multi-modality, we randomly choose one from the three nearest agents. However, during our experiments in Section 5.2~5.3, predictions are made for all agents.
>
> >The authors did not explicitly list any limitations. One possible limitation is the focus solely on illustrating static versus dynamic driving performance discrepancy.
>
> **Though we have discussed our limitation on planner selection and simulation scope in Section 6, we totally agree it is crucial to emphasize the limitation posed**. One common and critical factor is the asymmetry of prediction error. Even when two prediction models exhibit the same ADE, their driving performances will not be identical. This discrepancy arises from the different influence of prediction errors to planning. We will add this limitation in our final version.
>
> [1]Agarwal et al. Deep reinforcement learning at the edge of the statistical precipice. NeurIPS 2021.
>
> [2]Pillutla et al. Mauve: Measuring the gap between neural text and human text using divergence frontiers. NeurIPS 2021.
>
> [3]Hu et al. Planning-oriented autonomous driving. CVPR 2023.
>
> [4]Liang et al. Learning lane graph representations for motion forecasting. ECCV 2020.
>
> [5]Zhou et al. Query-Centric Trajectory Prediction. CVPR 2023.
>
> [6]Lin et al. The architectural implications of autonomous driving: Constraints and acceleration. ASPLOS 2018.
>
> [7]Yamaguchi et al. In-vehicle distributed time-critical data stream management system for advanced driver assistance. JIP 2017.

---

> > ### Comment · Reviewer_RCyQ · 2023-08-18
> >
> > Thank you for responding to my comments.
> >
> > I appreciate that this work may give a more comprehensive discussion of relevant metrics to indicate actual driving performance after training (static versus dynamic ADE, computational performance) but I still currently feel that the level of contribution and novelty is not quite sufficient for NeurIPS.
> >
> > I imagine static datasets will still be used in training for quite some time (ex: for real-world applications) and so I would of found it more useful if ways to address this misalignment were proposed.
> >
> > The proposed metrics do more strongly indicate actual driving performance, but they also require direct evaluation in an interactive simulation environment and pairing with a planner and so I am not sure of their usefulness as a final evaluation metric instead of simply using driving performance itself, although they can act as a useful tool to indicate why driving performance may be poor. But as a tool for indicating why driving performance may be poor, these again seem like obvious metrics where dynamic ADE is essentially the same prediction error computed during evaluation (a sort of test set error) and computation time is clearly relevant for real-time planners. I agree that closed-loop metrics are often overlooked when evaluating the motion prediction model but I am unsure if this is due to the fact that they are unknown or simply to do with the hurdles in integrating them into the evaluation pipeline: they require an interactive simulation environment (difficult for real-world data) and pairing with a planner from which results may vary from planner to planner. Or is it possible that since the prediction models are now being evaluated on different datasets as an outcome of their interaction with the environment, the comparison is no longer fair (i.e. one dataset is harder than the other)? Thus I am still somewhat unsure of the level of contribution of this work.

---

> > > ### Author Response · Authors · 2023-08-20
> > >
> > > We thank the reviewer for engaging in the discussion! Regarding concerns about the novelty of our paper, it is essential to emphasize that the dynamics gap is not a sort of test set error. **The dynamics gap is the inherent limitation of open-loop evaluation, in which exo-agents will not react to the change of predictor and corresponding ego-motion**. Conversely, in the real-world, different predictors result in varied behaviors of the ego-agent, which, in turn, influence the future behaviors of other road users, leading to different dynamics within the environment. Without addressing this gap, all prediction methods remain limited in real-world utility.
> > >
> > > While we agree that datasets will still be used for training purposes in the future, a shift towards dynamic datasets will happen, rather than relying on static datasets. Dynamic datasets (or offline simulators) like MixSim [8] and nuPlan [9] are promising to replace the role of current open-loop datasets. Researchers have the opportunity to train and evaluate their models using interactive agents, instead of static records. However, the awareness of the urgency to evaluate predictors within closed-loop remains limited. These dynamic datasets are all designed for evaluating planners. Besides, there is no comprehensive study about the limitation of current open-loop evaluation, and the key factor leading to it.
> > >
> > > Although dynamic datasets have yet to be employed for prediction evaluation, the reveal of dynamics gap can still make substantial contributions to the autonomous driving field. We present a potential application here: motion model evaluation.
> > >
> > > Motion Model Evaluation: While each simulator employs its unique motion model for exo-agents, the assessment of motion model quality remains challenging. Through the utilization of dynamic ADE, three values can be computed utilizing the same predictor and planner: static ADE within the simulation dataset, dynamic ADE within the simulation environment, and dynamic ADE during real-world testing. These metrics solely involve changes in the motion model of exo-agents, enabling the evaluation of the simulation's fidelity to real-world agents' motion. e.g.
> > >
> > > $$
> > > fidelity =\frac{ADE_{real} - ADE_{sim}}{ADE_{real} - ADE_{static}}
> > > $$
> > >
> > > Again, we would like to thank the reviewer for thoughtful and comprehensive comments. We would be more than glad to discuss any remaining concerns you might have.
> > >
> > >
> > > [8] Suo et al. MixSim: A Hierarchical Framework for Mixed Reality Traffic Simulation. CVPR 2023.
> > >
> > > [9]nuPlan Planning Challenging: Closed-loop reactive agents, 2023

---

> > > > ### Comment · Reviewer_RCyQ · 2023-08-20
> > > >
> > > > To clarify: When I likened the dynamic ADE to a "sort of test set error," I did not mean to imply an exact equivalence to a conventional static test set in the machine learning context. This was not the main point of my comment. I realize that evaluation in a closed-loop interactive environment after open-loop training is not a conventional fixed test set. It changes based on the trained motion prediction model paired with a planner and the reactions of other agents in the scene to the ego vehicle, and this introduces additional complexities. I mentioned this in my original reply.
> > > >
> > > > The purpose of my comment was to point out that using dynamic ADE as a tool to diagnose the dynamics gap seems like an obvious check to see if things are working correctly during evaluation (i.e. like thinking to measure the test set error). Given that these metrics (re-calculating ADE during closed-loop evaluation and computation time) seem relatively self-evident I feel this reduces the impact of the work. I also feel that the interplay findings between dynamic ADE and computation time affecting driving performance is also somewhat obvious (with computation time being more relevant to driving performance when the planning time budget is tight).
> > > >
> > > > If the main purpose of the work is also to make the performance gap issue known then I also feel that this lacks a high level of novelty since other works have also pointed out the discrepancies in performance that arise from open-loop training and closed-loop evaluation giving rise to out-of-distribution scenarios. I also question if the lack of closed-loop evaluation is always related to a lack of knowledge that this dynamics gap exists (that this work is trying to make known) or simply other hurdles in the evaluation pipeline instead. For example: closed-loop evaluation may not be possible for real-world datasets, or variability introduced when paired with different planners.
> > > >
> > > > I agree that addressing this performance gap is an important area of future research. But I am unsure if simply pointing out the issue (which already seems known from previous works) with (what I consider obvious) metrics to evaluate it meets the contribution requirements for NeurIPS.
> > > >
> > > > I feel the work has value: it is an important issue to discuss, experiments confirm the claims made, and it gives greater experimental detail as to why this gap occurs and have increased my contribution score slightly to reflect this. But I unfortunately feel that the contribution level is still not significant given my previous comments.

---

> > > > > ### Author Response · Authors · 2023-08-21
> > > > >
> > > > > We divide the questions into two aspects: **the awareness of dynamics gap**, and **the reason behind the absence of reactive evaluation research**.
> > > > >
> > > > > For the first question, it is mentioned that the dynamics gap appears rather self-evident, thus reducing the impact of our work. However, the 'self-evident' limitations of current prediction evaluation are numerous: awareness of drivable area, near collision, social conventions, etc. Nevertheless, quantitative analysis is missing to determine the priority of addressing these limitations and to find the dominant factor on ultimate driving performance.
> > > > >
> > > > > An illustrative instance is the asymmetry of prediction error [10]. Even when two prediction models demonstrate identical ADE values, their driving performances will not align due to the varying impact of prediction error on ego-plan. This factor may appear self-evident, and the challenge lies in assessing its relative significance. In our experiment, we account for the most critical aspect of asymmetry prediction error by comparing "Closest Dynamic ADE" with "Dynamic ADE." Consequently, the influence of asymmetry on the correlation between ADE and Driving Performance is **much smaller** than that of dynamic gap. (Influence For RVO: 0.16 vs 0.61; Influence For DESPOT: 0.20 vs 0.96)
> > > > >
> > > > > **Our contribution goes beyond mere revealing these factors. We offer a comprehensive quantitative analysis to uncover the dominant role of the dynamics gap in shaping real-world driving performance**. Unlike other factors, the development of prediction methods will encounter significant limitations if the dynamics gap remains unsolved.
> > > > >
> > > > > Upon addressing the first question, the response to the second question becomes evident: researchers have yet to recognize the significance of the dynamics gap. The common belief is that such a dynamics gap exerts marginal influence on evaluation and is not worth additional training and simulation to tackle. On the contrary, our experiments demonstrate that the dynamics gap holds a dominant influence on driving performance, which is the starting point of our paper.
> > > > >
> > > > > In addition, since the optimal solution to planning-aware prediction evaluation is to integrate it with a real planner, pairing with planners is not a drawback of our work. Considering the variation of planners may lead to concerns regarding fair prediction evaluation, a uniform cost function can be designed for all planners to form a fair comparison.
> > > > >
> > > > > As the discussion phase draws to a close, we would like to extend our gratitude once again for your prompt response and valuable insights.
> > > > >
> > > > > [10]Ivanovic, Boris, et al. "Injecting planning-awareness into prediction and detection evaluation." IEEE IV, 2022.

---

### Official Review · Reviewer_tS95 · 2023-07-07

**Soundness:** 3 good
**Presentation:** 3 good
**Contribution:** 2 fair
**Rating:** 5
**Confidence:** 3

**Summary:**

The paper provides an extensive study on the discrepancy between model prediction accuracy and driving performance. It explores two major factors: 1. The dynamics difference; and 2. The computational efficiency of predictors. Various methods are tested in two settings: 1. Fixed Number of Predictions; and 2. Fixed Planning Time. The paper evaluates the model performance on both open-loop prediction and closed-loop diving.

**Strengths:**

The experiments are well-designed and extensive, which provides strong evidence for the statements in the paper.

**Weaknesses:**

While the two factors are well supported by the experiment results given, it is insufficient to say whether other factors are less important, such as the data quality of the open loop eval, the unknowability (future events influencing the ground truth future trajectory) issue for open loop prediction, as well as multimodality in the diving scenarios.

**Questions:**

It would be great if the author could provide add some discussions on possible other factors that cause the discrepancy and why the two factors mentioned in the paper are dominant.

**Limitations:**

See questions.

---

> ### Author Rebuttal · Authors · 2023-08-09
>
> We appreciate your thoughtful analysis and constructive feedback on our paper. Your insights have been valuable, and we are grateful for the time you spent evaluating our work. Below, we address your main concerns:
>
> > While the two factors are well supported by the experiment results given, it is insufficient to say whether other factors are less important, such as the data quality of the open loop eval, the unknowability (future events influencing the ground truth future trajectory) issue for open loop prediction, as well as multimodality in the diving scenarios.
>
> Thank you for raising this insightful point. We acknowledge the importance of considering factors beyond the two emphasized in our study. Your comment has prompted us to further investigate the correlation between temporal consistency [1] and driving performance, focusing on how sudden changes in prediction might impact driving outcomes.
>
> Additionally, we have examined the relationship between prediction variance and driving performance to deepen our understanding of the influence of multimodality. The detailed findings from these analyses are provided in the PDF file, which will be available on August 9th "global response" . We believe that these additional investigations will contribute to a more comprehensive understanding of the complex interplay between these various factors.
>
> > It would be great if the author could provide some discussions on possible other factors that cause the discrepancy and why the two factors mentioned in the paper are dominant.
>
> We appreciate your suggestion to delve into other possible factors contributing to the observed discrepancies. One common and critical factor is the asymmetry of prediction error [2]. Even when two prediction models exhibit the same ADE, their driving performances will not be identical. This discrepancy arises from the different influence of prediction errors to planning. For instance, errors associated with the closest exo-agent can have a higher impact on the ego-agent's plan. The conflict between the predicted state of the exo-agent with the ego-agent's plan also matters.
>
> **However, it is important to note that the asymmetry of prediction error is not universal, which means that it occurs only in specific scenarios or for specific agents**. As a result, the influence of this factor is less significant than our proposed dynamic evaluation and computational efficiency, both of which impact prediction evaluation in every scenario and at each timestep. To provide quantitative evidence, we have effectively accounted for the most critical aspect of asymmetry prediction error by comparing "Closest Dynamic ADE" with "Dynamic ADE." Consequently, the influence of asymmetry on the correlation between ADE and Driving Performance is considerably less than the impact of dynamic evaluation. (Influence For RVO Planner: 0.16 vs 0.61; Influence For DESPOT Planner: 0.20 vs 0.96)
>
>
> [1] Ye, Maosheng, et al. “DCMS: Motion Forecasting with Dual Consistency and Multi-Pseudo-Target Supervision.” ArXiv.org, 27 Feb. 2023, arxiv.org/abs/2204.05859.
>
> [2] Ivanovic, Boris, and Marco Pavone. “Injecting Planning-Awareness into Prediction and Detection Evaluation.” IEEE Xplore, 1 June 2022, ieeexplore.ieee.org/document/9827101/.

---

### Official Review · Reviewer_Bibu · 2023-07-11

**Soundness:** 2 fair
**Presentation:** 2 fair
**Contribution:** 2 fair
**Rating:** 4
**Confidence:** 5

**Summary:**

This work attempts to look for evaluation metrics for trajectory prediction beyond the widely used ADE/FDE metrics. In particular, the authors focus on two overlooked aspects in prediction evaluation, which are: 1) the dynamics gap between the dataset and closed-loop driving scenarios; 2) the trade-off between computational efficiency and performance. The authors argue that we should turn to closed-loop prediction evaluation to address these two problems.

**Strengths:**

This work focuses on a very important and open question for trajectory prediction, which is looking for evaluation metrics for trajectory prediction beyond the widely used ADE/FDE metrics. It is good that the authors emphasize and validate the limitation of open-looped prediction accuracy and the importance of closed-loop evaluation for trajectory prediction.

**Weaknesses:**

1. It is good to see that the authors showed the limitation of open-looped evaluation and the importance of closed-loop evaluation in their experiments. However, I did not think the authors provided any new perspective or insightful solution to the problem of closed-loop evaluation. The value of closed-loop evaluation has been widely acknowledged in the autonomous driving community. It is one of the main motivations behind the emerging research on data-driven traffic simulation. The bottleneck is not the lack of prediction evaluation metrics under the closed-loop evaluation scheme, which is targeted by one of the two claimed contributions of the paper (i.e., dynamic ADE vs. static ADE), but the closed-loop simulation itself. How to synthesize a reliable simulator with realistic reactive agent behavior is an open question. We even do not have a commonly acknowledged set of metrics for evaluating simulation at this stage. Unfortunately, the authors did not discuss the impact of the simulation environment on the closed-loop evaluation result in this paper (e.g., whether the simulated driving performance and the simulated dynamic ADE metric are correlated with real-world driving performance). While the authors adopted different prediction models and planners in their evaluation, they only used a single simulator (i.e., SUMMIT). Moreover, the authors did not include a literature review on traffic simulation and evaluation metrics for traffic simulation.

2. The inherent logic behind the experiments is fundamentally flawed. In Sec. 5.2, the authors showed that static ADE is not correlated with dynamic ADE and driving performance. However, the authors used the correlation of static ADEs on the Argoverse and the Alignment dataset collected from SUMMIT as evidence to validate the realism of SUMMIT. Just like the authors' observation in Sec. 5.2, the correlation in static ADE does not necessarily indicate that the SUMMIT simulation is reliable and realistic under closed-loop simulation and evaluation. To justify the realism of SUMMIT, the authors should adopt those evaluation metrics from the latest literature and compare SUMMIT against those state-of-the-art simulation models.

**Questions:**

1. I do not think the authors appropriately answer the question raised in the title (i.e., what truly matters in trajectory prediction for autonomous driving?). The paper focuses on two perspectives on this question, which are very limited and far from compiling a comprehensive and insightful answer. The experimental setting is also insufficient as an attempt to answer such a bold question (e.g., lacking analysis on the impact of simulation on closed-loop evaluation).
2. It would be beneficial to have some qualitative analysis and visualization beyond statistics. For instance, visualizing some representative examples could be helpful in understanding why static ADE and dynamic ADE are not correlated.
3. The citation for SUMMIT is missing in the main text.

**Limitations:**

The authors discussed some limitations and future directions in Sec. 6. However, I think the limitations are not sufficiently addressed. For instance, I would consider the limited evaluation on a single simulator and the missing discussion on the impact of simulation realism as crucial drawbacks of this work.

---

> ### Author Rebuttal · Authors · 2023-08-09
>
> We thank the reviewer for the constructive comments! We kindly ask the reviewer to let us know if further clarification is needed.
> >I did not think the authors provide any new perspective or insightful solution. The value of closed-loop evaluation has been widely acknowledged in autonomous driving.
>
> We disagree with the question’s premise. While the value of closed-loop evaluation has been widely recognized, **its emphasis mainly centered on planning and control [7,8,9]**. It would be great if the reviewer can provide a pointer for prediction. Our work aims to shed light on the equally essential need for evaluating seemingly static modules, such as prediction, through closed-loop evaluation, which has been largely overlooked in most studies [1,2] and SOTA of competitions [3].
>
> >The bottleneck is not the lack of prediction evaluation metrics under the closed-loop evaluation, but synthesizing a reliable simulator with realistic reactive agent behavior.
>
> Not quite. As we have claimed in A1, the primary motivation behind this work is to draw attention to the importance of evaluating seemingly static prediction models with closed-loop simulation. We challenge the current evaluation system and propose a better evaluation protocol. The dynamic ADE is presented as an additional contribution. Synthesizing a real simulator falls beyond the scope of this research and suffices as a standalone research.
>
> >Unfortunately, the authors do not discuss the impact of the simulation environment… They only use a single simulator.
>
> **The impact of the simulation environment is not related to our core contribution**. To cope with realistic scenarios, the only way is to drive in the real-world, which is often unaffordable. Thus, simulators are usually taken as the proxy to carry out research. We would like to reiterate that our contribution is to point out that static prediction evaluation is not correlated with driving performance, which is stated in some papers [4,5], but is never adequately discussed and validated. We conduct extensive experiments with representative predictors and planners to support this claim. In addition, SUMMIT is built upon Carla, the widely-used simulator in recent competitions [6] and research [10,11].
>
> >Moreover, the authors did not include a literature review on traffic simulation and evaluation metrics for it.
>
> **Although traffic simulation is not relevant to our core contribution**, we agree that incorporating a comparison of simulators and explaining our choice of SUMMIT will strengthen our paper. We provide a literature review in the general response.
>
> >The inherent logic behind the experiments is fundamentally flawed… To justify the realism of SUMMIT, the authors should adopt those evaluation metrics against those SOTA simulation models.
>
> **The objective of this experiment is not to validate the realism of SUMMIT, but to ensure the consistency of prediction performance across different environments**. Even in different real-world datasets, the performance of the same predictor will differ. Thus, what we care about is predictors’ ranking of accuracy. We conduct the alignment experiment to demonstrate that the ranking of prediction metrics keeps stable across real-world dataset Argoverse and simulation dataset Alignment.
>
> >I do not think the authors appropriately answer the question raised in the title (i.e., what truly matters in trajectory prediction for autonomous driving?).
>
> We partially agree that our paper does not fully answer what truly matters in motion prediction. Our work focuses on proving that static evaluation does not correlate with driving performance, and advocates the superiority of dynamic evaluation. **What truly matters in motion prediction is a bold question to answer**. Expecting one paper to completely resolve this issue is unrealistic. Nevertheless, our work takes a stride forward, and our experiments fully support the current lack of prediction evaluation, leaving this question open for further exploration. With that being said, we concur that a more precise title would enhance the understanding of our main idea, and we will modify our title after further discussions.
>
> >It would be beneficial to have some qualitative analysis and visualization beyond statistics.
>
> We add the visualization for showing why Static ADE and Dynamic ADE are not correlated in the attached PDF. In static evaluation, the ground truth is determined based on dataset records and remains unaffected by predictors. However, in dynamic evaluation, different predictors result in varied behaviors of the ego-agent, which, in turn, influence the future behaviors of other road users, leading to different dynamics within the environment. This directly affects the ground truth of prediction as other agents behave differently, thus causing the disparity between Static ADE and Dynamic ADE.
>
> >The citation for SUMMIT is missing.
>
> Thank you for pointing it out. We will add the citation [12] in final version.
>
> [1]Hu et al. Planning-oriented autonomous driving CVPR 2023
>
> [2]Liang et al. Learning lane graph representations for motion forecasting ECCV 2020
>
> [3]Zhou et al. Query-Centric Trajectory Prediction. CVPR 2023
>
> [4]Ivanovic et al. Injecting planning-awareness into prediction and detection evaluation IV 2022
>
> [5]Ivanovic et al. Rethinking trajectory forecasting evaluation. arXiv 2021
>
> [6]NeurIPS CARLA Autonomous driving challenge, 2022
>
> [7]nuPlan Planning Challenging: Closed-loop reactive agents, 2023
>
> [8]Phan-Minh et al. Driving in Real Life with Inverse Reinforcement Learning. ArXiv 2023
>
> ‌[9]Cheng et al. MPNP: Multi-Policy Neural Planner for Urban Driving. IROS 2022
>
> [10]Danesh et al. LEADER: Learning Attention over Driving Behaviors for Planning under Uncertainty CoRL 2023
>
> [11]Ulfsjöö et al. On integrating POMDP and scenario MPC for planning under uncertainty–with applications to highway driving IV 2022
>
> [12]Cai et al. Summit: A simulator for urban driving in massive mixed traffic ICRA 2020

---

> > ### Comment · Reviewer_Bibu · 2023-08-13
> >
> > Thanks to the authors for their detailed response. Please find my follow-up comments to the response below:
> >
> > 1. Novelty: I agree with the authors that the closed-loop evaluation of trajectory prediction is a perspective that has not been directly studied in the literature. However, I still consider the work to have limited novelty, given that closed-loop evaluation has been studied for planning and control [7,8,9] and the limitation of open-loop evaluation has been stated in prior works [4, 5].
> >
> > 2. Simulation matters for closed-loop evaluation: I disagree with the author's statement that "the impact of the simulation environment is not related to our core contribution". The main difference between dynamic ADE and static ADE lies in evaluating the prediction model over state distribution under closed-loop driving. If the reactive agents' behavior in the simulation is not realistic in those OOD states, how could we trust the computed dynamic ADE metric? Thus, Discussing the effect of simulation to closed-loop evaluation of prediction models is a crucial aspect of the subject the authors aim to study. I understand the current set of results still have their value, but I don't think they are sufficient for a NeurIPS paper.
> >
> > 3. Realism of SUMMIT: In Lines 244-245 of the submitted manuscript, the authors claim that "The consistent results suggest that the Argoverse and Alignment datasets share similar underlying features. Therefore, the SUMMIT simulator can be employed to evaluate real-world performances." I think the second sentence does imply that the SUMMIT simulator is realistic. In fact, related to my last comment, I think evaluating the realism of SUMMIT is necessary in order to validate that the dynamic ADE computed in SUMMIT is reliable. If the authors agree that Fig. 2 cannot validate the realism of SUMMIT, then they should include additional results to justify the realism of SUMMIT.
> >
> > Overall, I still lean towards rejecting the paper in its current form.

---

> > > ### Author Response · Authors · 2023-08-15
> > >
> > > >Novelty: I agree ...However, I still consider the work to have limited novelty, given that closed-loop evaluation has been studied for planning and control [7,8,9] and the limitation of open-loop evaluation has been stated in prior works [4, 5].
> > >
> > > To address the concerns of novelty, we respond in three aspects.
> > >
> > > 1. While closed-loop evaluation has been explored extensively for planning and control [7, 8, 9], assuming its equivalent influence on static modules like predictions, without comprehensive experimentation,  is unjustified. The inadequacy of such studies has been acknowledged by reviewers as well. In this context, we present comprehensive analysis as the first to undertake this exploration, marking its novelty.
> > >
> > > 2. We stated that the limitations of the present prediction evaluation have been highlighted in prior works[4, 5].  However, as these papers state, they do not aim to address the inherent limitation of open-loop evaluation, but focus on the asymmetry of prediction error. Despite connecting prediction evaluation with driving performance, they still employ real-world datasets as the standard for evaluating predictors.
> > >
> > > 3. Conversely, we discover the dynamics gap between current prediction evaluation systems and real-world applications, which is overlooked in previous works. We substantiate this gap with experiments, and demonstrate its strong impact on the correlation between prediction performance and driving performance. As real-world tests are often unaffordable, an alternative approach–closed-loop evaluation– is suggested to ease the dynamics gap. The main contribution of our paper is to reveal such a dynamics gap and its significance, not the closed-loop itself.
> > >
> > > >Simulation matters for closed-loop evaluation: I disagree with the author's statement that "the impact of the simulation environment is not related to our core contribution"... Thus, Discussing the effect of simulation to closed-loop evaluation of prediction models is a crucial aspect of the subject the authors aim to study.
> > >
> > > Our results comprise two layers of analysis.
> > >
> > > The first layer establishes the existence and significance of the dynamics gap concerning the correlation between prediction performance and driving performance. The relationship between simulation datasets and simulation scenarios mirrors that of real-world datasets and real-world scenarios. Our investigation substantiates that the dynamics gap between datasets and real-time systems stands as a foundational factor contributing to the observed weak correlation between prediction evaluation metrics and real-time performance. Crucially, the dynamics gap can solely be effectively mitigated when the predictor is evaluated and applied within the same real-time environment, leading to a substantial correlation between prediction evaluation metrics and driving performance. Our core contribution lies in the analysis of the dynamics gap. This contribution remains unaffected by the realism of the simulator and necessitates real-world tests for the prediction evaluation.
> > >
> > > The second layer underscores the efficacy of the alternative evaluation protocol (simulation) when real-world tests are unaffordable. To substantiate this, we conduct the Sim-Real Alignment experiment, which serves to establish the alignment between prediction performance of the simulation dataset and real-world datasets. Likewise, the alignment of driving performance is verified via the simulator itself.  Thus, we take effort in identifying the optimal simulator, SUMMIT, which is built upon Carla, the most widely-used simulator in recent competitions [6] and research [10,11].
> > >
> > > The impact of the simulation environment partially affects the realism of the simulator and offers marginal influence to our contributions. Therefore, we opt not to engage in an in-depth discussion of it within the main text.
> > >
> > > >Realism of SUMMIT: In Lines 244-245 of the submitted manuscript, the authors claim that "The consistent results suggest that the Argoverse and Alignment datasets share similar underlying features. Therefore, the SUMMIT simulator can be employed to evaluate real-world performances.".... If the authors agree that Fig. 2 cannot validate the realism of SUMMIT, then they should include additional results to justify the realism of SUMMIT.
> > >
> > > As aforementioned, our core contribution revolves around the concept of the dynamics gap. The closed-loop simulation is served to address this gap only when real-world tests are unaffordable. Even if the simulator is not totally realistic, our core conclusion still stands. The realism of SUMMIT only supports part of our secondary contribution. Besides, although the requirement of realism of the simulator can be eased, we still tried our best to find the best simulator SUMMIT, which is built upon Carla, the most widely-used simulator in recent competitions [6] and research [10,11].

---

> > > > ### Comment · Reviewer_Bibu · 2023-08-15
> > > >
> > > > Thank the authors for their follow-up clarification. I do appreciate the authors' effort in identifying the existence and significance of the dynamics gap. I think it is a valid point and the major contribution of this work. However, I don't think this point itself is sufficient to make this paper accepted for NeurIPS, considering that closed-loop evaluation has been studied for planning and control [7,8,9] and the limitation of open-loop evaluation has been stated in prior works [4, 5].
> > > >
> > > > That's why I am suggesting an in-depth discussion on the effect of simulation on the closed-loop evaluation metrics. I agree that the closed-loop evaluation of the prediction model is a promising direction. However, it cannot serve as a sound solution for benchmarking prediction models if the simulation environment is not realistic, even though It could be an acceptable solution for engineering purposes. For example, if one model achieves a smaller dynamic ADE than the other in SUMMIT, does it necessarily mean it is a better prediction model than the other? Since SUMMIT leverages a data-driven prediction model to simulate the reactive agents' behavior, the "ground truth" used for computing the dynamic ADEs can be drastically wrong on OOD states, then the ranking based on dynamic ADE cannot be trusted in this case.
> > > >
> > > > After reading all the authors' responses, I am willing to change my rating to 4, which reflects my current impression of this work. Overall, I do think the paper has its value to some degree, but I don't think its contribution is sufficient when considered for NeurIPS.

---

> > > > > ### Author Response · Authors · 2023-08-18
> > > > >
> > > > > We are glad that some of your concerns have been addressed and thank you for raising the score! We would like to reiterate that prior works [4, 5] still focus on crafting planning-aware metrics in an open-loop context, instead of closed-loop scenarios. It is crucial to underscore that our contribution mainly stands on providing a fresh perspective to prediction evaluation, highlighting both the presence and equal significance of the dynamics gap and computational efficiency. We expect a highly enhanced correlation with real driving performance using our evaluation protocol, which is realized in our experiments.
> > > > >
> > > > > Since conducting real-world tests is expensive, researchers rely on simulators as an alternative. Simulators have been utilized to conduct numerous research and have achieved significant progress in autonomous driving, e.g.,  planning and control. In light of that, we opted to employ the CARLA-based simulator, SUMMIT, to enhance the reliability of our results. Thus, while we agree synthesizing a reliable simulator would undoubtedly enhance the credibility of our paper, the current simulator adequately serves to validate our conclusions and provides promising evaluation results. The proposed dynamic ADE notably surpasses current open-loop evaluation metrics, although it is not perfect.
> > > > >
> > > > > We extend our gratitude for your prompt and valuable responses, as well as the time you spent in our discussions.

---

### Official Review · Reviewer_Aygk · 2023-07-20

**Soundness:** 2 fair
**Presentation:** 2 fair
**Contribution:** 3 good
**Rating:** 4
**Confidence:** 3

**Summary:**

In the article, the author discusses the limitations of the current open-loop trajectory evaluation method: 1. It lacks consideration of closed-loop interactions with the ego-planner 2. It overlooks computational timeliness. These two factors result in a misalignment between open-loop trajectory evaluations and closed-loop driving scores. The author also proposes a new metric, Dynamic ADE/FED, which enables a positive correlation to be established between open-loop trajectory evaluations and closed-loop driving scores.

**Strengths:**

* Originality: This paper is original, providing a new evaluation method for the field of trajectory prediction.
* Significance: The foundation of this paper is sound. It focuses on analyzing the flaws and issues in the standards of trajectory prediction evaluation, and conducts a substantial amount of experiments across different models.
* Quality & Clarity: I believe this section is the weakness of the paper. Please refer to the 'Weakness' part.

**Weaknesses:**

* Clarity & Quality:
    - 1. The core Dynamic ADE/FDE metrics in this paper are not clearly explained; there is hardly any section dedicated specifically to elaborating on this aspect. It would be beneficial to include an illustration for better understanding.

    - 2. Despite the numerous experiments conducted, some of them fail to support the arguments made. For instance, on lines 243-245, you stated that the positive correlation demonstrated by different prediction models on the Argoverse and Alignment datasets indicates that there isn't a significant domain gap between the SUMMIT simulator and real-world data. I find this reasoning flawed. A positive correlation merely indicates consistency in the predictive ability of the models across the two datasets; it does not necessarily imply that the behaviors in the two datasets are consistent. Moreover, a closer look at the specific numerical values on the x and y-axes in Figure 2 reveals significant discrepancies between the Argoverse and Alignment datasets. The ADE distribution for Argoverse lies between 1.5-3.5, while that for Alignment falls between 0.5-1.5. This suggests that the Argoverse dataset is substantially more challenging than the Alignment dataset, which contradicts your claim.
    - 3. Besides this, there are many other similar issues. The overall experimental conclusions lack substantial empirical support and are simply based on statistical analyses of a small number of data points, yielding correlations that aren't particularly strong (the paper only employs eight different prediction models and two different planning models). I remain unconvinced by statistical conclusions drawn from such a limited set of data points.
* Novelty:
    - 4. This paper seems more akin to an experimental report. Despite the multitude of experiments conducted, they remain rather superficial. For a NeurIPS paper, a deeper level of analysis is required, such as quantitative analysis, comparing what specific interactions in which scenarios cause differences between static and dynamic ADE in open-loop and closed-loop evaluations. Additionally, as I previously mentioned, many of the experimental results do not effectively support the conclusions drawn.
    - 5. The work done in the 'Computational Efficiency' section is quite trivial. The conclusions drawn here are rather obvious, and in the field of autonomous driving engineering, this aspect has already been thoroughly considered.

**Questions:**

Please refer to the 'weaknesses' section.

**Limitations:**

Please refer to the 'weaknesses' section.

---

> ### Author Rebuttal · Authors · 2023-08-09
>
> Thanks for your thoughtful comments! We kindly ask the reviewer to let us know if further clarification is needed.
>
> >The core Dynamic ADE/FDE metrics are not clear… It would be beneficial to include an illustration.
>
> This is a great point for us to clarify more! We add a figure in the attached PDF. Similar to static ADE, dynamic ADE is also calculated by the average L2 distance between the forecasted trajectory and the ground truth. However, in dynamic evaluation, different predictors result in varied behaviors of the ego-agent, which, in turn, influence the future behaviors of other road users, leading to different dynamics within the environment. This directly affects the ground truth of prediction as other agents behave differently.
>
> >Some experiments fail to support the arguments… A positive correlation does not necessarily imply that behaviors in the two datasets are consistent.
>
> **The argument we support is the consistency of prediction performance across Argoverse and Alignment datasets**. By guaranteeing this, real-world prediction performance is provided for experiments. Real-world behavior is a means to gather actual driving performance, as conducting real-world tests is often unaffordable, we employ SUMMIT to conduct our research. The SUMMIT simulator, built upon Carla, a widely-used simulator in recent competitions [4] and research [5,6], offers behavior approximating reality.
>
> >The ADE for Argoverse lies … while that for Alignment lies… Argoverse is more challenging than Alignment.
>
> The prediction results for a given predictor can vary, even when applied to two real-world datasets. However, such variations do not necessarily imply that one dataset is more challenging than the other. (SOTA of nuScenes: minADE5=1.092m in 6s prediction horizon, SOTA of Waymo: minADE6=0.535m in 8s prediction horizon) Nevertheless, the underlying behavior of drivers should be similar across real-world datasets. The observed differences can be influenced by many factors, e.g. average speed, number of agents, or other unidentified variables. **We argue that if the accuracy ranking of predictors remains consistent (as indicated by ADE/FDE alignment), the prediction ability of models are aligned, enabling us to evaluate predictors with SUMMIT**.
>
> >Conclusions lack solid empirical backing, relying on limited data points for statistical analysis, resulting in weak correlations.
>
> **We want to clarify that we have conducted extensive experiments to support our claim**. Specifically, we collected 50 scenarios for each predictor in each setting, resulting in a total of 1600 simulation scenarios. To ensure clarity of results, we only utilize the average performance for each predictor, and the correlation to driving performance is strong. **Additionally, the incorporation of a predictor or planner is expensive**. Assuming a predictor requires 0.02s to execute once, the DESPOT planner calls the predictor 1000 times at each step to explore adequate tree nodes, leading to 20s per planning step. Given the planner's requirement to operate at 3Hz, we set tick rate to 1Hz, allowing for 10s real time for each planning. Each scenario comprises around 200 steps, the total runtime for one predictor sums up to 200\*50\*10/3600=28 days. As many predictors are even much slower than 0.02s (e.g., KNN: 0.224s), it is infeasible to exhaust all possible predictors given the computational cost, not to mention the implementation time required.
>
> >Despite the multitude of experiments conducted, they remain rather superficial.
>
> **This paper shed light on the equally essential need for evaluating seemingly static modules, such as prediction, through closed-loop evaluation**, which has been largely overlooked in most studies [1,2] and SOTA of Argoverse Competition [3]. In addition, we reveal the **trade-off between computational efficiency and accuracy even when the predictor is fast enough for the planner's execution. We are the first to provide a comprehensive investigation on these problems**. We further analyze the correlation between other two possible factors and driving performance to support our core research in the attached PDF.
>
> >The work done in Computational Efficiency is trivial. The conclusions are obvious and have been thoroughly considered.
>
> The important idea we want to point out is **the trade-off between computational efficiency and accuracy even when the predictor is fast enough for the planner's execution**, which has been neglected by most research. In the context of autonomous driving, a consensus exists that the whole perception system must meet a specified threshold for proper planning. Once this threshold is satisfied, the focus shifts to ensuring accuracy. (e.g., 100ms [7,8]) However, our experiments demonstrate a notable trade-off that persists much below the specified threshold. When the predictor runs much faster than the threshold (2ms), the driving performance still remains dominated by the fastest prediction method (CV). Only with an ample planning time budget (Tick Rate = 1Hz), methods with better accuracy dominate. This trade-off holds crucial importance, as various planners exhibit unique trade-offs in addition to their fundamental requirement.
>
> [1]Hu et al. Planning-oriented autonomous driving. CVPR 2023
>
> [2]Liang et al. Learning lane graph representations for motion forecasting. ECCV 2020
>
> [3]Zhou et al. Query-Centric Trajectory Prediction. CVPR 2023
>
> [4]NeurIPS CARLA Autonomous driving challenge, 2022
>
> [5]Danesh et al. LEADER: Learning Attention over Driving Behaviors for Planning under Uncertainty. CoRL 2023
>
> [6]Ulfsjöö et al. On integrating POMDP and scenario MPC for planning under uncertainty–with applications to highway driving. IV 2022
>
> [7]Lin et al. The architectural implications of autonomous driving: Constraints and acceleration. ASPLOS 2018
>
> [8]Yamaguchi et al. In-vehicle distributed time-critical data stream management system for advanced driver assistance. JIP 2017

---

> > ### Comment · Reviewer_Aygk · 2023-08-18
> >
> > Thank you for the further elaboration. Some of my concerns have been resolved.
> >
> > I would like to clarify my previous question further:
> > 1. Regarding the consistency issue between Argoverse and Alignment datasets: My intention was not to say that SUMMIT cannot be employed to evaluate real-world performances, but rather to say that your experiment does not adequately reflect your claim. Your designed experiment only shows that different models have positively correlated trajectory prediction performance on these two datasets, which doesn't prove that SUMMIT has the same behavior with the real world scenario. It is possible that SUMMIT and real-world datasets have significant differences in performance, but due to factors intrinsic to the model itself (such as the amount of model parameters, structure, etc.), they show positive correlations in prediction metrics across different datasets. An experiment that could successfully validate your claim might be to demonstrate the consistency of agent behavior in SUMMIT with that of Argoverse agents, such as constructing a SUMMIT scenario that mimics the Argoverse dataset, and then calculating the consistency of agent trajectories in that scenario.
> >
> > Additionally, I have some questions about the starting point of the paper:
> >
> > 1. Since dynamic ADE/FDE requires closed-loop evaluation, why don't we directly use a closed-loop Planning and Control metric, such as Driving Score?
> > 2. Closed-loop evaluation of dynamic ADE/FDE indeed exhibits better consistency with closed-loop metrics, but we can only test it in a closed-loop simulator. This metric cannot be computed on real-world data.
> > 3. Moreover, this metric is influenced by the planner involved in the evaluation; different planners would lead to significant inconsistencies in the dynamic ADE/FDE measurements of trajectory prediction models, as referenced in Figure 4.
> >
> > These factors could have a significant impact on the practical utility of the dynamic ADE/FDE metric.

---

> > > ### Author Response · Authors · 2023-08-20
> > >
> > > To answer these questions, we would like to explain more about the starting point of the paper. Our paper comprises two layers of analysis:
> > >
> > > The first layer establishes the existence and significance of the dynamics gap and computational efficiency, concerning the correlation between prediction performance and driving performance. **The relationship between Alignment dataset and SUMMIT simulator mirrors that of Argoverse dataset and real-world autonomous driving, as each system pair shares identical ego-planner and exo-agents motion model.** The difference in the correlation coefficient between dynamic ADE/FDE and static ADE/FDE versus driving performance is notable, demonstrating the significance of the dynamics gap. Similarly, the importance of computational efficiency is emphasized by the remaining gap between dynamic ADE and driving performance. These two gaps can solely be effectively mitigated when predictors are evaluated in real-world autonomous driving. This layer constitutes our core contribution.
> > >
> > > The second layer emphasizes the efficacy of the alternative evaluation protocol (simulation) when real-world tests are unaffordable. **To substantiate this, we conduct the Sim-Real experiment, which establishes the alignment of prediction performance between simulation and real-world. Likewise, the alignment of driving performance is verified via the simulator itself.**  SUMMIT is built upon Carla, the most widely-used simulator in recent competitions [4] and research [5,6]. The attached motion model GAMMA [9] responds to ego-motion and outperforms that of Carla. Afterward, the ranking of dynamic metrics reflects the ranking of real-world driving performances and is more obtainable.
> > >
> > > To ensure the protocol's effectiveness across diverse planners, we employ the RVO planner to explore the correlation between dynamic ADE and driving performance. Significantly, a strong relationship between the two persists.
> > >
> > > >Regarding the consistency between Argoverse and Alignment datasets: I'm saying that your experiment does not adequately reflect your claim.
> > >
> > > As aforementioned, the Sim-Real experiment ensures alignment in prediction performance, while the alignment of driving performance (the same as behavior) is verified by the SUMMIT simulator along with its affiliated motion model GAMMA [9]. Given our primary focus is not the development of a simulator, our efforts are dedicated to finding the best simulator.
> > >
> > > >Why don't we directly use a closed-loop Planning and Control metric rather than dynamic ADE/FDE?
> > >
> > > We agree that the ultimate goal is driving performance. The subtle difference behind using dynamic ADE lies in emphasizing the significance of the dynamics gap. We select a prediction metric that calculates in the same way but exhibits distinct performance in static and dynamic evaluations. In addition, computational efficiency is emphasized by the remaining gap between dynamic ADE and driving performance.
> > >
> > > The underlying question lies in the significance of dynamic metrics. As aforementioned, our main contributions are: 1. establish the existence and significance of the dynamics gap and computational efficiency. 2. emphasize the efficacy of the alternative evaluation protocol. It's essential to note that dynamic ADE serves solely as the tool we employ to substantiate these contributions, rather than constituting our primary contribution.
> > >
> > > >We can only test closed-loop metrics in a closed-loop simulator. This metric cannot be computed on real-world data.
> > >
> > > As aforementioned, the relationship between Alignment dataset and SUMMIT simulator mirrors that of Argoverse dataset and real-world. Thus, dynamic metrics also prove effective in real-world tests. However, they are unattainable from datasets. This underscores our primary finding regarding the importance of real-time evaluation.
> > >
> > > >Moreover, this metric is influenced by the planner involved in the evaluation...influence the practical utility of the dynamic metrics.
> > >
> > > This concept forms one cornerstone of our paper: prediction evaluation is dependent on the downstream modules. Previous planning-aware metrics [10] also tried to capture the impact, but in an open-loop manner. It is inappropriate to posit that the 'optimal' predictor will excel 'normal' predictors across all planners. The best predictor for a given planner might vary, depending on its real driving performance. This, naturally, leads to corresponding changes in the ranking of dynamic ADE. Therefore, our main contributions are: 1. identifying the importance of dynamics gap and computational efficiency 2. highlighting the efficacy of the alternative evaluation protocol, rather than proposing the dynamic ADE.
> > >
> > > We extend our gratitude to the valuable insights and constructive feedback, which greatly enhance the quality of our manuscript.
> > >
> > > [9]Luo et al. Gamma: A general agent motion model for autonomous driving
> > >
> > > [10]Ivanovic et al. Injecting planning-awareness into prediction and detection evaluation

---

### Author Rebuttal · Authors · 2023-08-09

We sincerely thank all the reviewers for your constructive feedback and recognition of this work. The reviewers found:

* The problem is “original” (R-Aygk) and “interesting” (R-1Qcn).
* The experiments are “well-done” (R-tS95, R-RCyQ, R-1Qcn) and  “extensive” (R-tS95, R-RCyQ).
* The work provides “an excellent study setup” (R-1Qcn) and “highlights the flaws in the widely used metrics of trajectory prediction evaluation,” (R-1Qcn, R-Aygk).

We would like to re-emphasize this work's technical contributions:
* We shed light on the equally essential need for evaluating seemingly static modules, such as prediction, through closed-loop evaluation.
* We reveal the trade-off between computational efficiency and accuracy even when the predictor is fast enough for the planner's execution.
* We propose Dynamic ADE as a new evaluation metric in the closed-loop evaluation.

We have made the following claims more clear according to all the reviewers’ insightful comments.
* **The purpose of Sim-Real Alignment (Section 5.1)**. Even in different real-world datasets, the performance of predictors will differ. (SOTA of nuScenes: minADE5 = 1.092m in 6s, SOTA of Waymo: minADE6 = 0.5345m in 8s) Thus, what we care about is predictors’ ranking of accuracy. As a result, we conduct the alignment experiment to demonstrate that the ranking of prediction performance keeps stable across real-world dataset Argoverse and simulation dataset Alignment.
* **The detailed definition of Dynamic ADE and its difference to Static ADE**. Similar to Static ADE, Dynamic ADE is also calculated by the average L2 distance between the forecasted trajectory and the ground truth. However, in closed-loop evaluation, different predictors result in varied behaviors of the ego-agent, which, in turn, influence the future behaviors of other road users, leading to different dynamics within the environment. This directly affects the ground truth of prediction as other agents behave differently, as explained in the attached PDF.
* **The trade-off between computational efficiency and accuracy**. A consensus exists that the perception system must meet a specified threshold (e.g. 100ms) for proper planning. Once it is satisfied, the focus shifts to ensuring accuracy. However, our experiments demonstrate a notable trade-off even when the predictor runs considerably faster than the threshold (2ms). Only in experiments with ample planning time budget (Tick Rate = 1Hz), the method with better accuracy dominates.

We also include more analysis to support our ideas:
* Other possible factors that may cause the discrepancy and why the two factors mentioned in the paper are dominant.
* Exploring the correlation between temporal consistency and driving performance to investigate whether sudden changes in prediction (mainly occur when encountering unknown future issues) impact driving performance.
* Examining the correlation between prediction variance and driving performance to gain a more profound understanding of the influence of multimodality.

We add the related works to make it clearer why we choose SUMMIT:
* Several rule-based simulators are designed for autonomous driving systems. CARLA [1] offers a range of sensors and agent types but relies on predefined maps and exhibits relatively low density with simple rule-based behaviors. Another simulator [2] supports real-world maps, yet its rule-based planner complicates prediction performance evaluation.
* Recently, learning-based simulators [3,4] have emerged, generating diverse and realistic behaviors from actual data using open-source datasets. However, these simulators are restricted to predetermined maps. Generating realistic behaviors for new maps may prove challenging.
* SUMMIT [5] stands out by offering rich-context urban maps, realistic visuals, and intricate traffic behavior. Agent behaviors are generated using GAMMA [6], an advanced multi-agent motion prediction model. Leveraging SUMMIT's advanced capabilities, we can provide a comprehensive assessment with various planners and predictions.

We also add the equation for the combination of performance metrics to the main text, following the valuable comment of reviewer RCyQ:

$$
\bar{P}_{\textrm{metrics}}=\begin{cases}
\frac{P\_{\textrm{metrics}} - P^{\textrm{min}}\_{\textrm{metrics}}}{P^{\textrm{max}}\_{\textrm{metrics}} - P^{\textrm{min}}\_{\textrm{metrics}}}, & \textrm{metrics} = \\{\textrm{efficiency}\\} \\\\
1-\frac{P\_{\textrm{metrics}} - P^{\textrm{min}}\_{\textrm{metrics}}}{P^{\textrm{max}}\_{\textrm{metrics}} - P^{\textrm{min}}\_{\textrm{metrics}}}, & \textrm{metrics} = \\{\textrm{safety}, \textrm{comfort}\\}
\end{cases}
$$
where $P^{\textrm{min}}\_{\textrm{metrics}}$ and $P^{\textrm{max}}\_{\textrm{metrics}}$ represent the minimum and maximum pairs of each performance metric among all scenarios.

We kindly ask the reviewers to let us know if further clarification or information is needed.

[1] Dosovitskiy et al. CARLA: An open urban driving simulator CoRL 2017

[2] Lopez et al. Microscopic traffic simulation using sumo ITSC 2018

‌[3] Caesaret et al. NuScenes: A Multimodal Dataset for Autonomous Driving CVPR 2020

‌[4] Igl et al. Symphony: Learning Realistic and Diverse Agents for Autonomous Driving Simulation ICRA 2022

[5] Cai et al. Summit: A simulator for urban driving in massive mixed traffic ICRA 2020

[6] Luo et al. Gamma: A general agent motion model for autonomous driving RAL 2022

---

### Decision · Program_Chairs · 2023-09-21

**Decision:**

Accept (poster)

**Comment:**

The paper investigates for trajectory prediction the discrepancy between performance on fixed datasets and the driving performance when applied to downstream applications. It proposes two overlooked factors that might affect the broader impact and yet less well-studied in the community.

All reviewers reach concensus that the paper is well written; the formulation is easy to follow and the problem setup worth investigating. The experiments have been conducted effectively to verify the key claims. The final rating is 4,4,4,5,7. The main concerns by the negative comments are:

- "the effect of simulation on the closed-loop evaluation". The proposed metric (e.g. dynamic ADE) does not imply a good indicator for a proposed method.

- the paper looks more like a survey / detailed report of existing methods.

Authors did a detailed rebuttal. There are quite a few back and forth dicussions among reviewers and authors. Since this is a borderline paper, AC steps in, read the paper, rebuttal, author feedback and discussions carefully. The final recommendation is Accept based on:

- The paper did actually observe key factors that impede the motion prediction community: discrepancy between fixed dataset and real scenarios (e.g. interactive with other agents in the enviroment). The observations found in the paper are **highly relevant as great reference for the general audience**.

- As mentioned by Reviewer 1Qcn, the paper opens a discussion on how to align the discrepancy between open-loop (fixed dataset) and closed-loop (realistic scenario) metric for planning. This could serve as the preliminary step to trigger more investigation on this direction.

As such, despite there are some flaws (e.g. experiments on simulation rather than real-world) in the paper, AC believes that these are another story or fixable, and trust authors (based on rebuttal) could polish the manuscript to great extent. It is highly recommended to consolidate all comments, open-source the project and make results reproducible in a timely manner.